# Global surveillance of antimicrobial resistance in food animals using priority drugs maps

Cheng Zhao [1], Yu Wang[1], Ranya Mulchandani [1] &
Thomas P. Van Boeckel [1,2,3] ✉

Antimicrobial resistance (AMR) in food animals is a growing threat to animal health and potentially to human health. In resource-limited settings, allocating resources to address AMR can be guided with maps. Here, we mapped AMR prevalence in 7 antimicrobials in *Escherichia coli* and nontyphoidal *Salmonella* species across low- and middle-income countries (LIMCs), using 1088 point-prevalence surveys in combination with a geospatial model. Hotspots of AMR were predicted in China, India, Brazil, Chile, and part of central Asia and southeastern Africa. The highest resistance prevalence was for tetracycline (59% for *E. coli* and 54% for nontyphoidal *Salmonella*, average across LMICs) and lowest for cefotaxime (33% and 19%). We also identified the antimicrobial with the highest probability of resistance exceeding critical levels (50%) in the future (1.7–12.4 years) for each 10 × 10 km pixel on the map. In Africa and South America, 78% locations were associated with penicillins or tetracyclines crossing 50% resistance in the future. In contrast, in Asia, 77% locations were associated with penicillins or sulphonamides. Our maps highlight diverging geographic trends of AMR prevalence across antimicrobial classes, and can be used to target AMR surveillance in AMR hotspots for priority antimicrobial classes.

Antimicrobials are life-saving drugs used to treat infections in humans. However, the majority (73%) of antimicrobials sold globally are used in animals raised for food[1]. In animals, antimicrobials are used for treatment but also as surrogates for good hygiene practices and to increase productivity on farm in some regions of the world[2]. Antimicrobials have facilitated the intensification of animal farming, and enabled meeting a growing demand for animal proteins worldwide. From 2000 to 2020, In Brazil and China – the largest exporter and importer of meat among low- and middle- income countries (LMICs), meat production has grown by 89% and 23% respectively[3]. However, in LMICs, during the same period, the percentage of antimicrobials with prevalence of resistance higher than 50%[4] rose from 15% to 41% in chicken, and from 13% to 34% in pigs, with important consequences for animal health, and potentially for human health[5,6].

In high-income settings such as the US[7], Canada[8], and the EU countries[9], animal AMR has been the focus of systematic surveillance for decades. Surveillance data have supported policies that helped limiting the use of certain classes of antimicrobials in animals (e.g. third-generation cephalosporins such as ceftiofur[8]). However, in LMICs, systematic surveillance remains at best nascent, and point prevalence surveys (PPS) have been used as surrogates to systematic surveillance to infer regional trends in AMR in animals[10]. Thus far, these attempts at documenting trends in AMR using PPS relied on summary metrics such as the fraction of antimicrobials tested in a survey with prevalence of resistance higher than 50% (P50). For LMICs, trends in AMR have not yet been disaggregated for individual antimicrobial-bacteria combinations. This is a major limitation for potentially taking targeted actions on individual antimicrobial classes. One such action was the 2005 ban of

[1]Health Geography and Policy Group, ETH Zürich, Zürich, Switzerland. [2]One Health Trust, Washington DC, USA. [3]Spatial Epidemiology Lab, Université Libre de Bruxelles, Brussels, Belgium. ✉e-mail: thomas.van.boeckel@gmail.com

fluoroquinolones in poultry in the United States that was supported by surveillance data of fluoroquinolone-resistant *Campylobacter*[11]. For humans, systematic reviews—including in LMICs—helped estimate the global burden of AMR for 88 individual antimicrobial-bacteria combinations[12]. Conducting a symmetrical exercise for animals would enable a more targeted approach to the management of AMR in animals, and also comparison with patterns of AMR in humans[13].

The World Health Organization's list of Medically Important Antimicrobials (MIA)[14] is a natural starting point for developing drug-specific guidelines for surveillance of AMR in animals, and define priorities for actions. However, the MIA list does not explicitly account for considerable geographic variations of AMR levels within countries such as Kenya, China, Thailand where subsistence farming and industrial farming co-exist, and where access to veterinary services varies considerably between regions. In regions with high AMR levels, first line antimicrobials for disease treatment may have lost efficacy. Having the ability to predict which antimicrobials will cross critical resistance levels in the future could help assess the risk of antimicrobial resistant infections acquired from animal sources, as well as strengthening local surveillance effort. To the best of our knowledge, there are currently no maps that prioritize antimicrobial classes for surveillance in animals based on local epidemiological patterns of AMR. This is largely due to the lack of fine-grained geographic information on AMR prevalence for individual antimicrobial-bacteria combinations in systematic surveillance systems. PPS conducted at individual locations provide a unique opportunity for supplementing these efforts, and mapping priority antimicrobials for AMR surveillance. However, several challenges must be addressed to transform data extracted from event-based surveillance (PPS) into actionable epidemiological information. Firstly, combinations of antimicrobial-bacteria vary between PPS, and a panel of combinations that are abundantly represented across PPS must be selected to ensure comparability. Secondly, not only local environmental and anthropogenic covariates but also patterns of co-resistance between antimicrobials observed in PPS[15,16] can be informative of the future resistance profiles, but an appropriate computational framework must be developed to transform these statistical associations into predictions of classes of antimicrobials that will reach critical resistance levels.

In this study, we used 1088 PPS to map, at $10 \times 10$ kilometer resolution, the prevalence of resistance to 7 antimicrobials in *Escherichia coli* and nontyphoidal *Salmonella* species in food animals. We combined the maps of resistance prevalence with environmental and anthropogenic covariates as well as patterns of co-resistance to predict, in each location, which antimicrobials had the highest probability of exceeding critical levels of resistance (10%, 25% or 50%) in the near future. Our output is a global map displaying fine-scale variations of these drugs that will reach critical resistance levels, and could serve a basis to refine AMR surveillance efforts across regions.

## Results

### Trends of AMR
The mean prevalence of resistance weighted by the number of samples in each PPS, in *E. coli* and nontyphoidal *Salmonella*, was respectively 59% ($n = 745$) and 54% ($n = 597$) for tetracycline (TET), 57% ($n = 779$) and 46% ($n = 632$) for ampicillin (AMP), 45% ($n = 649$) and 36% ($n = 501$) for sulfamethoxazole-trimethoprim (SXT), 35% ($n = 656$) and 26% ($n = 553$) for chloramphenicol (CHL), 30% ($n = 796$) and 26% ($n = 624$) for ciprofloxacin (CIP), 28% ($n = 882$) and 23% ($n = 650$) for gentamicin (GEN), and 33% ($n = 446$) and 19% ($n = 334$) for cefotaxime (CTX). Between 2000 and 2019, changes in the prevalence of resistance were +12% (TET), +33% (AMP), +19% (SXT), +20% (CHL), +16% (CIP), +11% (GEN), and +37% (CTX) (Fig. 1). The temporal increases of resistance were significant ($p < 0.05$) for all antimicrobials apart from TET.

Prevalence of resistance was investigated in poultry in 52% ($n = 570$) of PPS, in cattle in 38% ($n = 409$) of PPS, and in pigs in 28%

($n = 303$) of PPS. Prevalence of resistance increased significantly for AMP, CHL, CIP, and CTX for poultry, and for AMP, SXT, CHL, CIP, GEN, and CTX for pigs (Supplementary Figs. 1 and 2). However, temporal trends of resistance were not significant for any antimicrobial classes for cattle (Supplementary Fig. 3).

We used ensemble geospatial modeling to map the prevalence of resistance to 7 antimicrobials in *E. coli* and nontyphoidal *Salmonella* in animals (Methods). In *E. coli*, resistance hotspots defined as N50 ≥ 3 were predicted in southern and eastern China, central Asia, northern India, northern Brazil, and Chile (Fig. 2h). In nontyphoidal *Salmonella*, resistance hotspots were predicted in northeastern China (Fig. 3d). Maps of resistance using other cutoff values (N20 and N35) were shown in Supplementary Fig. 7. Northern and eastern Brazil was also resistance hotspots for AMP resistance in *E. coli* (Fig. 2b). Northeastern China was resistance hotspots for SXT and GEN resistance in *E. coli*, as well as CHL, CIP, and GEN resistance in nontyphoidal *Salmonella* (Figs. 2c, f, 3d, e, f). Uncertainty of the predictions was the highest for CTX resistance in nontyphoidal *Salmonella* and the lowest for TET resistance in nontyphoidal *Salmonella* –standard deviation of predictions across all pixels on the map was on average 19.9% and 4.5%, respectively (Supplementary Figs. 8 and 9).

### Mapping priority antimicrobials for AMR surveillance—where and for which antimicrobial class will resistance prevalence exceed critical levels in the future?
We used risk factors (Supplementary Table 3) associated with the locations of each resistance profile reported in PPS, in combination with histories of acquisition of resistance phenotypes, to map which antimicrobials had the highest probability of its resistance prevalence exceeding critical levels (10%, 25% or 50%) in the future (Methods). This resulted in global maps of 'priority antimicrobials' for AMR surveillance. Using 50% as the critical level of resistance prevalence, the predicted priority antimicrobials were TET or AMP in 78% locations In Africa and South America (Fig. 4a). In contrast, in Asia, 77% locations were associated with AMP or SXT, because resistance to TET has already exceeded 50% in the vast majority of locations (83%). Concretely, SXT was the priority antimicrobial in northeastern India, southern and northeastern China, southern Brazil, Turkey, and Iran; AMP was the priority antimicrobial in northern and western China, Mongolia, and western India. In southern and eastern China, CHL was the predicted priority antimicrobial. Predictions of GEN and CTX having the highest probability of exceeding 50% resistance were not frequent (0.02%) and scattered in Asia and South America. The uncertainty associated with the predicted priority antimicrobials was on average 12% across all pixels (Fig. 4b), and was high (> 40%) in parts of western Brazil, South Sudan and North Korea. The percentage of pixels with high uncertainty (> 40%) for each country was calculated in Supplementary Table 11. We estimated the time for resistance prevalence to exceed 50%, for the predicted priority antimicrobial in each $10 \times 10$ km pixel (Supplementary Fig. 10). Across locations where AMP was the predicted priority antimicrobial (Supplementary Fig. 10b), the average time weighted by animals' biomass was 1.7 years, while for CIP the average time was 12.4 years (see Supplementary Table 4 for the average estimated time for each antimicrobial class). Maps of priority antimicrobials using 10% and 25% as critical levels of resistance prevalence are shown in Supplementary Fig. 7.

We assessed the accuracy of the predicted priority antimicrobials using spatial cross-validation (Methods) and calculated the area under the receiver operating characteristic curve (AUC) for predicting the probability of resistance prevalence exceeding 50% for each antimicrobial. The AUCs ranged from 0.880 to 0.994 between antimicrobials. We calculated the influence of each covariate for explaining the divergence in prediction accuracy by sequentially excluding these covariates from the models, and calculated the loss in

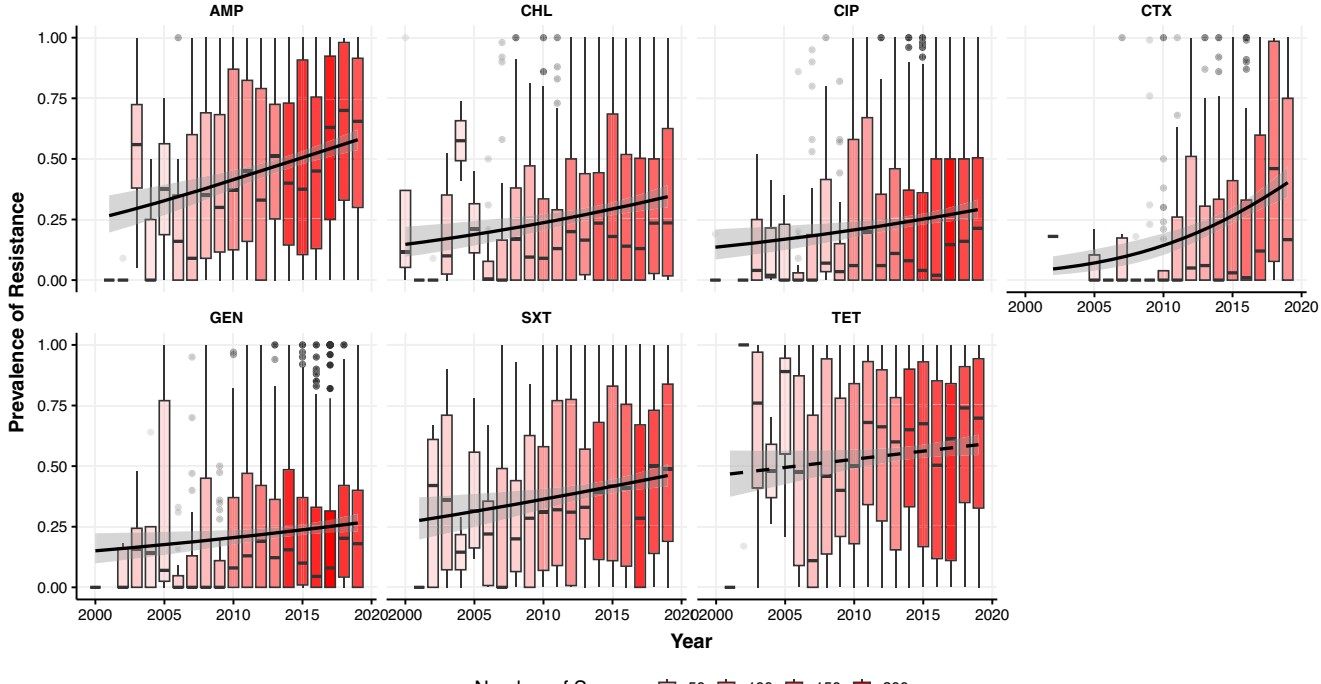

**Fig. 1 | Temporal trends of the prevalence of resistance in low- and middle-income countries for ampicillin (AMP), chloramphenicol (CHL), ciprofloxacin (CIP), cefotaxime (CTX), gentamicin (GEN), sulfamethoxazole-trimethoprim (SXT), and tetracycline (TET).** Transparency levels of the red colors are proportional to the number of surveys published each year. The 25th and 75th percentiles are represented by the lower and upper limits of each box, and the median value is marked with a horizontal line. Lengths of whiskers are 1.5 times the inter-quartile ranges, and values outside of this range are shown as individual points. Logistic regression is used to fit temporal trends of resistance prevalence. Solid lines represent significant temporal trends ($p < 0.05$; AMP: $p = 0.00000043$, CHL: $p = 0.00057$, CIP: $p = 0.0043$, CTX: $p = 0.00000023$, GEN: $p = 0.023$, SXT: $p = 0.0050$), and dashed lines represent nonsignificant trends (TET: $p = 0.061$). No adjustments are made for multiple comparisons. The 95% confidence intervals of the estimated temporal trends are shown in the gray areas.

AUC. Co-resistance patterns had the highest influence on predicting resistance to all antimicrobials, with ΔAUC ranging from 0.224 to 0.494. In contrast, environmental and anthropogenic covariates had limited added value for predicting whether resistance exceeds 50% in TET and AMP (ΔAUC 0.002 and −0.003), yet they increased prediction accuracy for other antimicrobials (ΔAUC ranging from 0.109 to 0.416). Covariates that were most frequently associated with the probability of resistance prevalence exceeding 50% were antimicrobial use, pesticide application rate, tri-annual cycles of precipitation, and amplitudes of night land surface temperature (Supplementary Table 5).

## Discussion

In this study, we mapped the distribution of resistance prevalence for 7 antimicrobials in *E. coli* and nontyphoidal *Salmonella* in food animals in low- and middle-income countries. We mapped the antimicrobials with the highest probability of their resistance prevalence exceeding critical levels (10%, 25% or 50%) in the future.

### Geographic distribution of AMR

The predicted maps of AMR based on the number of antimicrobials with resistance higher than 50% (N50) were consistent with previous global estimates of AMR in Van Boeckel & Pires et al. 2019. This consistency can be partly attributed to the incorporation of a subset of PPS used in Van Boeckel & Pires et al. 2019 into the present analysis (Supplementary Information). In both analyses, China, Turkey, Iran, India and Brazil were identified as hotspots of AMR. However, the previous authors estimated trends of AMR for four pathogens combined. Our analysis was conducted for *E. coli* and nontyphoidal *Salmonella* separately, and ensured comparability for monitoring AMR trends by including data on 7 drugs each representing a medically important class of antimicrobials. Our choice of proxies was also in line

with other global surveillance initiatives, such as the Global Tricycle Surveillance that uses ESBL-producing *E. coli* as the proxy[17].

In this analysis, we showed that the geographic distribution of AMR varied depending on the bacteria considered. For example, Iran was resistance hotspots of penicillins and amphenicols resistance in *E. coli* but not in nontyphoidal *Salmonella*. On average, *E. coli* had higher prevalence of resistance compared with nontyphoidal *Salmonella* for all antimicrobials. We also showed that the geographic distribution of AMR varied depending on the classes of antimicrobials considered. For example, in either *E. coli* or nontyphoidal *Salmonella*, northeastern China was identified as resistance hotspots for all antimicrobials except tetracyclines and penicillins. These two classes of antimicrobials have already reached high levels of resistance globally, leaving the preservation of the other antimicrobials of particular importance. Therefore, this region may need intensified policy intervention to contain AMR. Despite variations of AMR trends between antimicrobials, there were also consistencies on their geographic distributions. For example, Africa had consistently lower AMR prevalence compared the rest of the world for all antimicrobials, possibly because it consumes the least amount of veterinary antimicrobials compared with the rest of the world[18].

The 7 antimicrobial classes included in the analysis are the most frequently cited classes across 1,088 point prevalence surveys, and are important for treating infectious diseases in food animals. For example, tetracycline is widely used for treating Mycoplasma in chicken[19], gentamicin is used for treating *Pseudomonas aeruginosa* infections[20], and third- and fourth-generation cephalosporins are used for treating cattle mastitis[20]. Therefore, rising resistance levels in these drugs may lead to therapy failure, and thereby negatively impact animal health and the agricultural economy. Measures to contain AMR in the identified hotspot regions will need to be focused on reducing

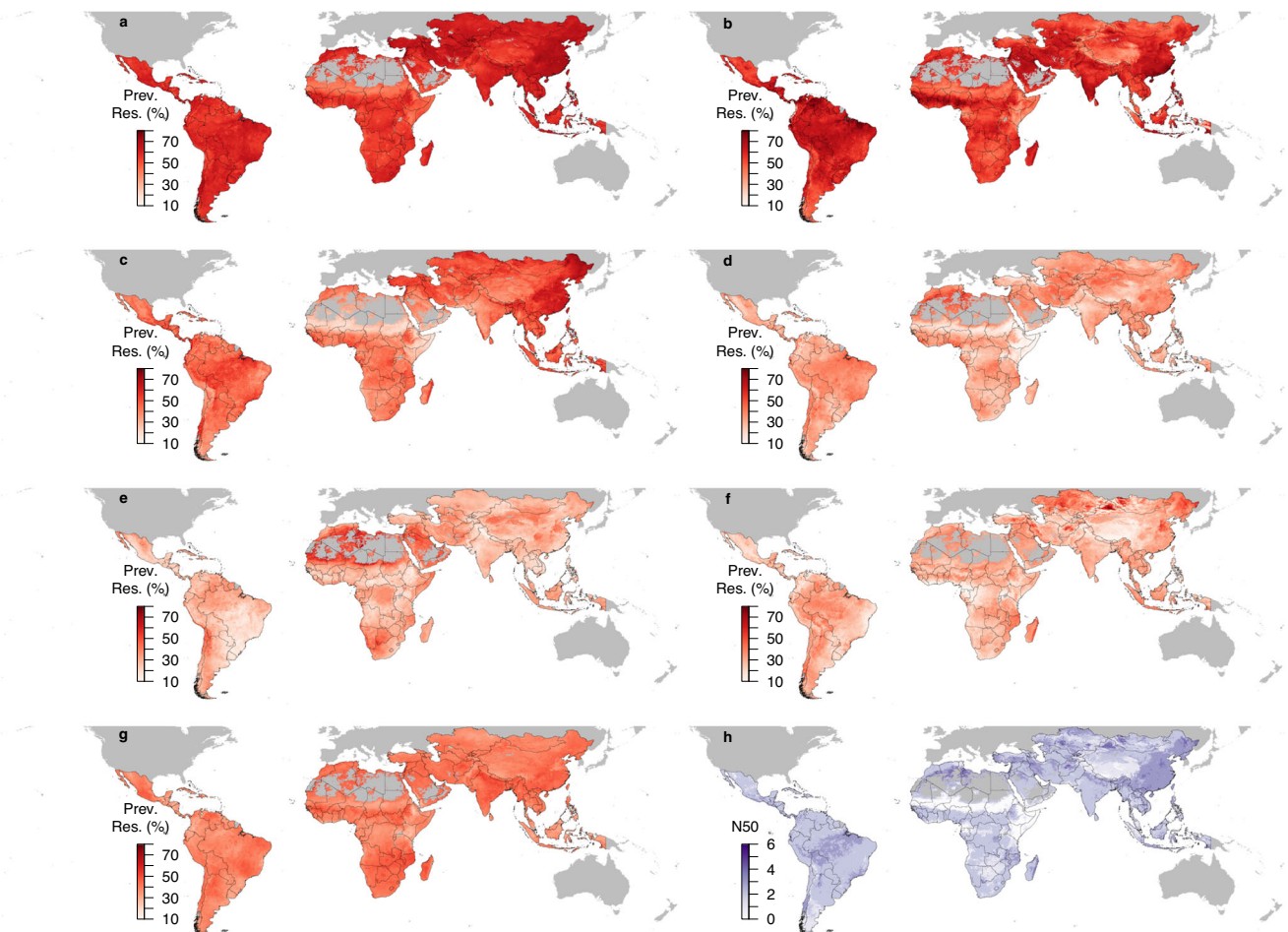

**Fig. 2 | Geographic distribution of antimicrobial resistance in *E. coli* in low- and middle-income countries between 2000 and 2019 (median year 2015).** Prevalence of resistance (Prev. Res.) for tetracycline **a**, ampicillin **b**, sulfamethoxazole-trimethoprim **c**, chloramphenicol **d**, ciprofloxacin **e**, gentamicin **f**, cefotaxime **g**. Overall resistance level across antimicrobials measured using the number of antimicrobials (out of 7) with resistance higher than 50% (N50; **h**) (See Supplementary Fig. 7 for maps generated using cutoff values other than 50%). Maps of resistance prevalence for the 7 antimicrobials are available on resistancebank.org.

antimicrobial use as well as strengthening biosecurity in farms. Enforcing a regulation with a cap of 50 milligram antimicrobial used per kilogram of food animal products was estimated to reduce global antimicrobial consumption by 64%[1]. However, major investment on the surveillance of antimicrobial use is needed for such regulations to be effective. Improving biosecurity in farms may reduce the reliance on antimicrobials for keeping the animals healthy. Measures to improve biosecurity include stricter hygienic control on farm entry and better separation between compartments in the farm, and can be facilitated by risk-based quantitative tools[21].

## Priority antimicrobials for AMR surveillance

We developed a computational approach (Methods), to map priority antimicrobials for surveillance that incorporates dependencies on local risk factors such as antimicrobial use and animal density, as well as history of acquisition of resistance phenotypes in one location. This approach uses spatial variations of resistance profiles of multiple drugs to infer which antimicrobial has the highest probability of its resistance prevalence exceeding critical levels (10%, 25% or 50%) in the future. If 50% was considered the critical level of resistance prevalence, in regions with currently low resistance levels (N50 = 0 or 1), tetracyclines and penicillins were the most frequently predicted antimicrobials with their resistance prevalence exceeding 50% in the future. Predictions of these two antimicrobial classes were based primarily on patterns of co-resistance between antimicrobials, with little influence from environmental and anthropogenic covariates. This suggested that such

patterns were universal across regions with low AMR, with the following possible reasons. Firstly, tetracyclines are among the cheapest and most accessible antimicrobials globally[22]. Secondly, tetracycline and ampicillin were discovered the earliest among the 7 antimicrobials included in the analyses. Their routine application for growth promotion in farms started as early as in the 1950s[23]. These factors may make them drugs of choice for application in food animals in regions with limited budgets and where their resistance has not yet been established[24].

In contrast, in regions with high AMR levels (N50 ≥ 2), sulfonamides and amphenicols were the antimicrobials with the highest probability that their prevalence of resistance will exceed 50% in the future. For amphenicols, the predictions were in eastern and southern China, where resistance to tetracyclines, penicillins, and sulfonamides were already above 50%. In China, despite chloramphenicol being banned for use in food animals since 2002 and other amphenicols being banned as growth promoters in 2020, increases in the prevalence of resistance to chloramphenicol[25] and florfenicol[26] continued to be observed years after the restrictions took place. The increases may be caused by the continued use of the drugs despite changes in regulation, or by co-selection of their resistance (e.g., associated with class 1 integrons) due to the use of other drugs such as dihydrostreptomycin and trimethoprim[27]. Our predictions suggested that future surveillance on use of amphenicols and its resistance could be intensified in these regions to better understand mechanisms underlying these trends.

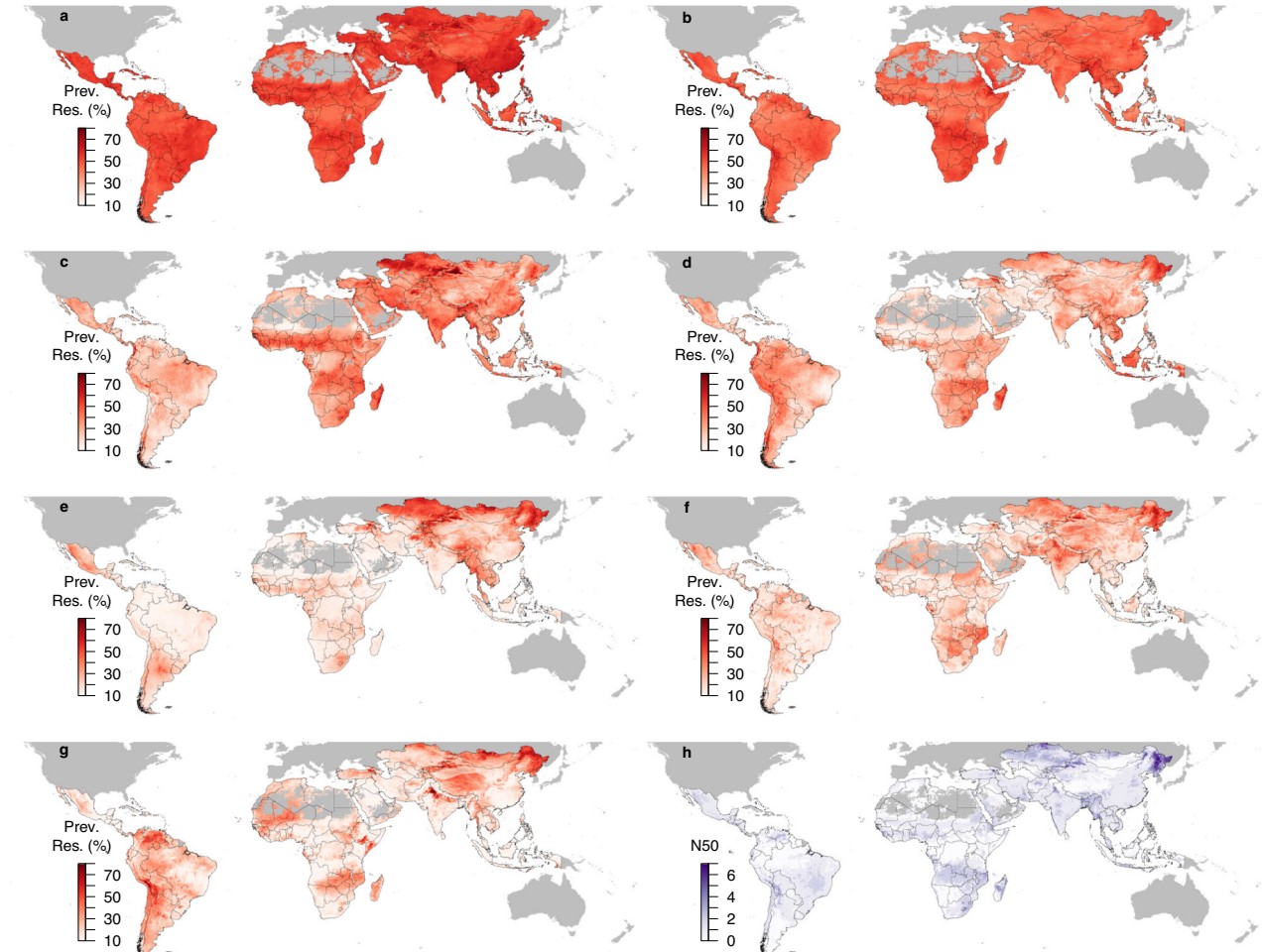

**Fig. 3 | Geographic distribution of antimicrobial resistance in nontyphoidal _Salmonella_ in low- and middle-income countries between 2000 and 2019 (median year 2015).** Prevalence of resistance (Prev. Res.) for tetracycline **a**, ampicillin **b**, sulfamethoxazole-trimethoprim **c**, chloramphenicol **d**, ciprofloxacin **e**, gentamicin **f**, cefotaxime **g**. Overall resistance level across antimicrobials measured using the number of antimicrobials (out of 7) with resistance higher than 50% (N50; **h**) (See Supplementary Fig. 7 for maps generated using cutoff values other than 50%). Maps of resistance prevalence for the 7 antimicrobials are available on resistancebank.org.

Environmental and anthropogenic covariates were predictive of the priority antimicrobials for AMR surveillance, particularly in regions associated with high levels of AMR. Population densities of animals were influential covariates, possibly because commonly applied antimicrobials differ between animal species[28]. Therefore, the difference in antimicrobial use across animal species may lead to difference in AMR. Temperature may affect the prevalence of animal injuries and therefore the frequency of (preventive) drug application[29].

We estimated the time it may take for resistance prevalence of the predicted priority antimicrobials to exceed a critical level. For locations where tetracyclines, penicillins or cephalosporins were the predicted priority antimicrobials, the average time for resistance to reach 50% across locations was below 7 years. Given that the median year of publication of the PPS was 2015, this implies that resistance may have already exceeded 50% now at these locations. For locations where amphenicols or quinolones were the predicted priority antimicrobials, their prevalence of resistance was estimated to exceed 50% in 2026 and 2027 on average across locations. However, the temporal trends of AMR used for estimating the time was based on data across low- and middle-income countries, and may differ depending on the geographic region. Estimating separate trends for each region is challenged by the limited amount of data in America and Africa countries, with large uncertainty associated with the estimated coefficients (Supplementary Table 2). Future work may be able to make region-

specific projections of AMR as the number of PPS published each year steadily increases and more data becomes available.

Our prediction of priority antimicrobials was based on surveys conducted exclusively on commensal _E. coli_ and nontyphoidal _Salmonella_ from healthy animals, and the majority of surveys used human clinical breakpoints to determine resistance phenotype. However, our approach could also be adapted to databases of AMR of other animal pathogens using veterinary clinical breakpoints, to help inform veterinarians on possible treatment options in regions of high AMR levels.

## Co-resistance between antimicrobials

Across surveys ($n = 1088$), resistance prevalence was significantly correlated between antimicrobials. All correlations were positive, a finding consistent with studies that interpreted collateral resistance using Markov network[30]. However, these observations were based on resistance profiles at the population level, rather than at the strain level where a diversity of both collateral resistance and sensitivity have been shown in silico[31] and in vitro[32]. Our results based on the amalgamation of PPS suggested that, at the population level, higher resistance in one drug is consistently associated with higher resistance in other drugs.

The highest correlations of resistance between antimicrobials were observed for sulfamethoxazole-trimethoprim and chloramphenicol, and for sulfamethoxazole-trimethoprim and tetracycline.

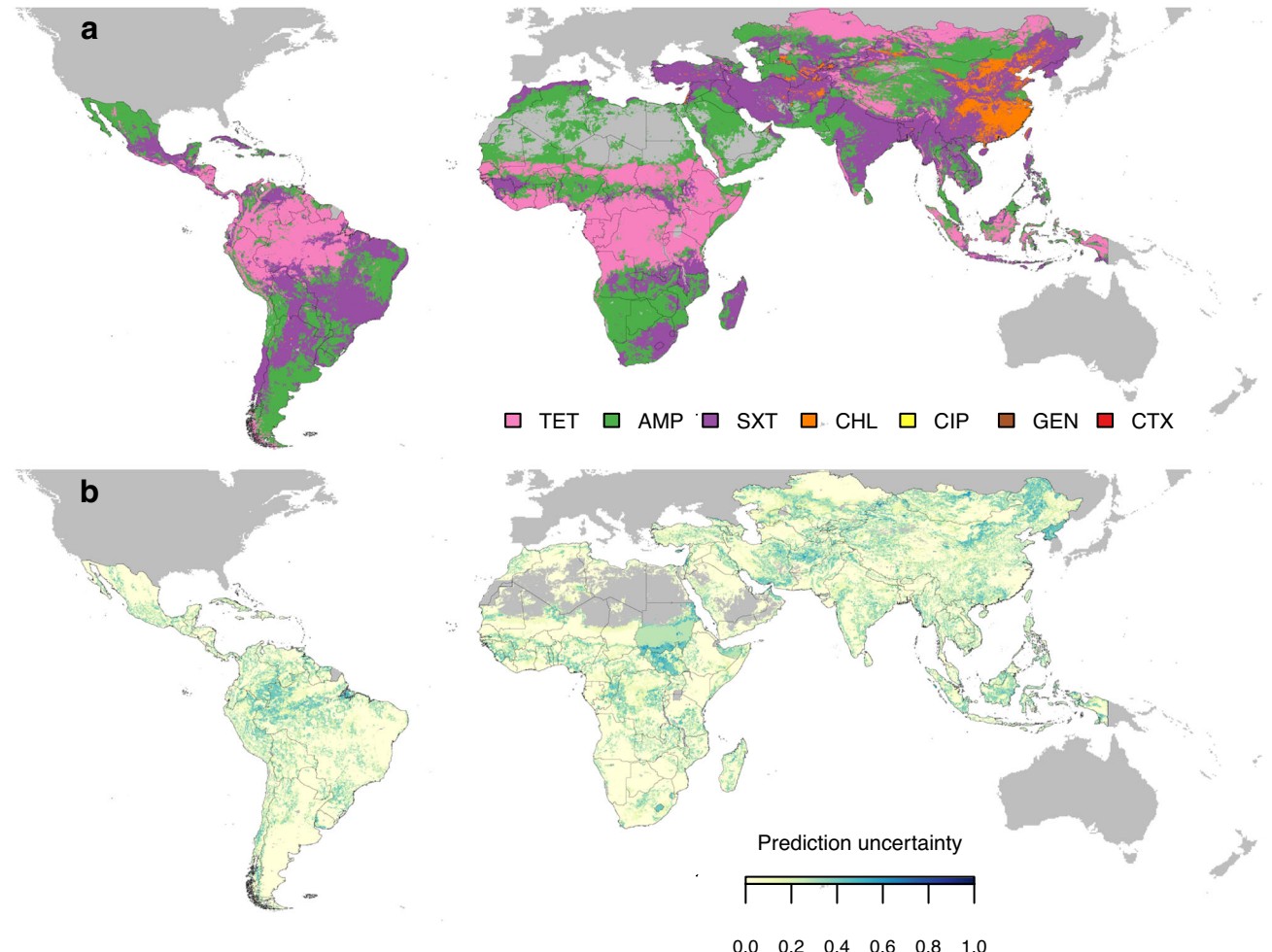

**Fig. 4 | Geographic distribution of priority antimicrobials. a** Geographic distribution of antimicrobials with the highest probability of their resistance prevalence exceeding 50% in the future in low- and middle-income countries. TET: tetracycline; AMP: ampicillin; SXT: sulfamethoxazole-trimethoprim; CHL: chloramphenicol; CIP: ciprofloxacin; GEN: gentamicin; CTX: cefotaxime. **b** Estimated uncertainty of the predictions shown in panel a, introduced by the imputation of missing resistance prevalence in the input dataset. Blues shades indicate the proportion of Monte Carlo simulations of imputed datasets, which generated different predictions compared with panel a (Methods).

One reason could be the co-location of several resistance genes on the same genetic element. For example, in *E.coli* isolated from pigs, chloramphenicol resistance gene *cmlA* was found on large plasmids that were linked to sulphonamide resistance genes *sul1* or *sul3*[33]. In addition, animals may often be exposed simultaneously to tetracyclines and sulfonamides, as these are antimicrobials the most frequently used in food animals[18].

## Limitations

As with any modeling study, our analysis comes with limitations. Firstly, predictive maps, as well as the imputation of missing resistance prevalence for modeling priority antimicrobials introduces uncertainty. The number of imputations was highest for cefotaxime—its resistance prevalence was missing in half (51%) of the surveys. However, the uncertainty of the missing values was captured by the high standard deviation (24%) of the multiple imputed values for cefotaxime. We attempted to quantify the uncertainty by combining Monte Carlo simulations of the imputed input datasets, and the variance of the Bayesian posterior predictive distribution for each simulation (Methods). Secondly, due to the limited number of surveys reporting resistance prevalence for individual antimicrobial-bacteria combinations, mapped predictions of AMR were restricted to 7 drugs and 2 bacteria. These drugs were amongst the most frequently used antimicrobial classes and the most frequently cited classes across 1088

point prevalence surveys. Additionally, predictions of nontyphoidal *Salmonella* were not disaggregated for individual serovars. However, this is in consistency with Murray et al. 2022 who mapped AMR in humans[12]. The limited number of surveys available also made it challenging to conduct spatio-temporal modeling, and we pooled together surveys from all years for AMR mapping. As the number of point prevalence surveys[34] published each year is growing, future efforts to map AMR may incorporate more antimicrobial-bacteria combinations and investigate both spatial and temporal effects on AMR maps, while insuring statistical robustness in the extrapolations. Thirdly, the maps of priority antimicrobials were built under the assumption that resistance prevalence will increase at the same rate as in the past 20 years, implying that the drivers behind AMR—including policies regulating antimicrobial use (AMU)—will remain unchanged in the near future. However, due to temporal changes in these policies—e.g. a 30% decline in antimicrobial use in Thailand from 2017 to 2019[18], the drivers behind AMR patterns may change in the future. Our predictions were intended to show how resistance may evolve without interventions on AMU policies, for the purpose of guiding such interventions. Fourthly, due to the lack of a systematic inventory of country-specific regulations on antimicrobial use, we did not explicitly include these regulations as covariates. For example, ciprofloxacin is banned in poultry in the US[11], but not in China[35]. However, the regulations were implicitly considered in the modeling process, with the inclusion of AMU in 2013 and 2020

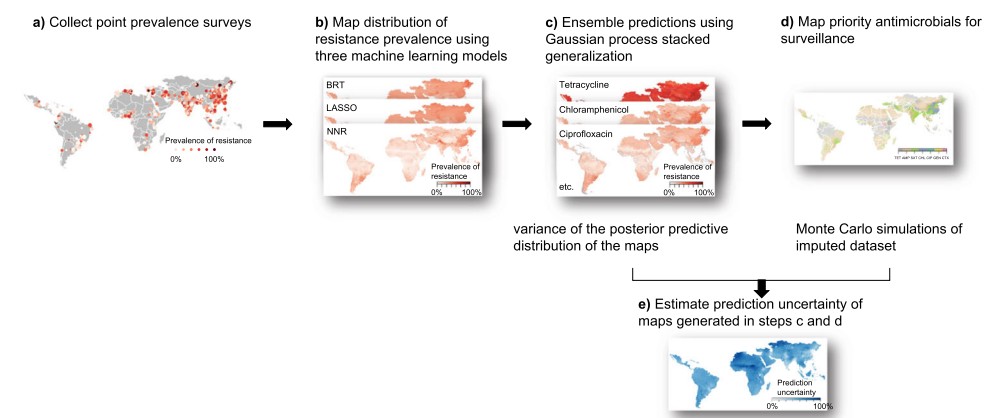

**Fig. 5 | Modeling framework. a** Collect point prevalence surveys. **b** Map distribution of resistance prevalence using three machine learning models: boosted regression trees (BRT), LASSO logistic regression (LASSO), feed-forward neural network (NNR). **c** Ensemble predictions using Gaussian process stacked generalization. **d** Map priority antimicrobials for surveillance. **e** Estimate prediction uncertainty of maps generated in steps **c** and **d**.

for each antimicrobial class as covariates. In 2013, Maron and colleagues reviewed restrictions on antimicrobial use in food animals[36]. However, to the best of our knowledge, an up-to-date global database on antimicrobial use policies has not been conducted. Fifthly, we dichotomized resistance prevalence using 50% threshold to define priority antimicrobials for AMR surveillance. We conducted sensitivity analysis by mapping priority antimicrobials using other thresholds (10% and 25%) as well. However, the choice of thresholds is dependent on multiple factors and in its nature subjective.

The maps of AMR produced in this study helps outline priorities for action. Firstly, in AMR hotspots—including China, Iran, India, Brazil and Chile, measures should be taken to further ongoing efforts to reduce antimicrobial use in food animals. Secondly, our analyses showed how AMR for 7 medically important antimicrobials may evolve in the future, without policy interventions. This could provide a baseline scenario where revisions of AMR policy could be based. Thirdly, the 3rd Global High-Level Ministerial Conference on AMR has set out a global target to reduce antimicrobials used in agrifood systems by 30–50% by 2030. Our maps could serve as a reference for more targeted measures aimed at specific antimicrobial classes in their corresponding hotspot regions of resistance. Possible measures include stricter regulations and on-farm monitoring on antimicrobial use, targeted awareness campaigns among veterinarians and farmers, as well as investments on improving farm hygiene to reduce dependence on antimicrobials.

## Methods

This analysis is structured in five steps (Fig. 5a–e): (a) collection and extraction of epidemiological information from point prevalence surveys (PPS); (b) mapping distribution of resistance prevalence using three machine learning models; (c) ensembling predictions using Gaussian process stacked generalization; (d) mapping priority antimicrobials for surveillance; and (e) estimating prediction uncertainty of maps generated in steps c and d. The literature review was conducted using Zotero (version 5.0.96.2) and Microsoft Excel (version 16.53), and all data analysis was conducted using R (version 4.1.1).

### Data collection and imputation

We extracted 1088 point prevalence surveys on AMR of *E. coli* and nontyphoidal *Salmonella* in healthy food animals across low- and middle-income countries (LMICs) across two decades between 2000 and 2019 (Supplementary Table 3). These surveys were collected through three rounds of literature review of four databases (PubMed, Scopus, ISI Web of Science, and China National Knowledge Infrastructure). The process of data extraction is explained in detail in the

Supplementary Information section "Literature review and data extraction". These surveys were conducted on major food animal species including cattle ($n = 409$), pigs ($n = 303$), poultry ($n = 570$), sheep ($n = 89$), horse ($n = 2$), and goat ($n = 2$). The animal samples used to determine resistance prevalence were taken from their meat (34% of total resistance prevalence), swabs from living animals on farm or in wet markets (32%), food products such as milk and eggs (16%), swabs from slaughtered animals (9%), and fecal samples on farm (7%).

In each survey, we extracted information on resistance prevalence, method used for antibiotic susceptibility testing (AST), guideline document used for performing AST, breakpoints used for assessing AST results, sample origin, number of animal samples and bacterial isolates, as well as the geographic location and time of the survey. The majority (91%) of the studies used the performance standards for antimicrobial susceptibility testing developed by the Clinical and Laboratory Standards Institute (CLSI) or the European Committee for Antimicrobial Susceptibility Testing (EUCAST). Each performance standards set breakpoints to classify resistance phenotypes, which are updated annually. These variations in breakpoints were adjusted using methods developed by Van Boeckel and colleagues[4], to maximize comparability between surveys.

For this analysis, we focused on 7 antimicrobial drugs: tetracycline (TET), ampicillin (AMP), sulfamethoxazole-trimethoprim (SXT), chloramphenicol (CHL), ciprofloxacin (CIP), gentamicin (GEN), and cefotaxime (CTX). The resistance prevalences of these drugs were the most frequently reported for their individual antimicrobial classes in the collected surveys (Supplementary Table 6), and therefore ensured robustness in comparisons made between surveys. These antimicrobial classes were all classified as critically important in veterinary medicine[20], and were also classified as either critically important or highly important for human medicine[14]. For each of the 7 drugs, we used all PPS that reported its resistance prevalence individually to map its distribution, with methods explained in the next sector. However, the subsequent prediction of priority antimicrobials requires complete resistance profiles with resistance prevalence of all 7 drugs. Therefore, 806 PPS that reported resistance prevalence of at least 4 out of these 7 drugs were included for this part of analysis. For the unreported antimicrobials, we imputed their resistance prevalence based on correlations between antimicrobials, using multivariate imputation by chained equations[37] (MICE; Supplementary Methods; Fig. 5a). The MICE algorithm imputed plausible values for 21% out of 9,877 antimicrobial resistance prevalence estimates in these surveys, while also providing a mechanism for integrating the uncertainty of imputation in the following analysis, as explained in section "Uncertainty".

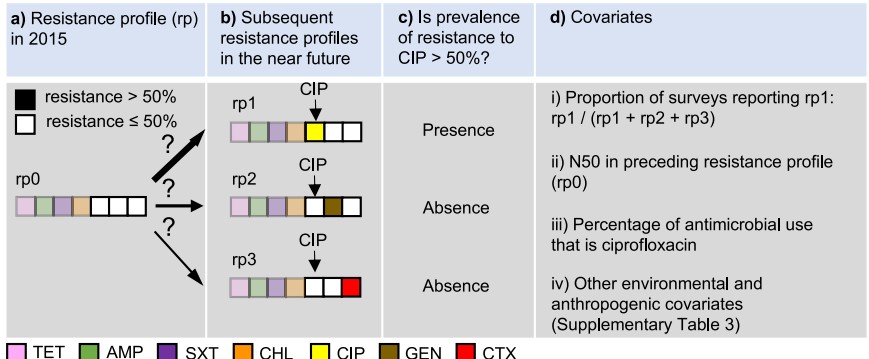

**Fig. 6 | LASSO logistic regression model to predict the probability that resistance prevalence of ciprofloxacin (CIP) will exceed 50% in the future, in pixels with predicted resistance profile (rp) of [1,1,1,1,0,0,0] (rp0) in 2015. a** Resistance profile in 2015. **b** Subsequent resistance profiles in the near future. **c** Determine whether prevalence of resistance to CIP is above 50%. **d** Covariates, including (i) the proportion of point prevalence surveys reporting the resistance profile in which resistance prevalence of ciprofloxacin exceeds 50% (rp1), out of all alternative antimicrobials (gentamicin in rp2, and cefotaxime in rp3), (ii) the number of antimicrobials with resistance above 50% (N50) in the predicted resistance profile in 2015 (rp0), (iii) the percentage of antimicrobial use (kg) of ciprofloxacin, and (iv) a set of environmental and anthropogenic covariates. TET tetracycline, AMP ampicillin, SXT sulfamethoxazole-trimethoprim, CHL chloramphenicol, CIP ciprofloxacin, GEN gentamicin, CTX cefotaxime.

## Trends of AMR for each antimicrobial class

We used logistic regression models to estimate temporal trends of resistance prevalence between 2000 and 2019 for each antimicrobial. For TET and AMP, we removed one outlier (DOI of PPS: 10.1264/jsme2.2000.173) out of 758 PPS reporting resistance for TET and 797 PPS reporting resistance for AMP, to ensure that the assumption of linearity between the logit of dependent variable and the independent variable was met based on results of Box-Tidwell test.

We mapped the distribution of the prevalence of resistance for each antimicrobial at 10 × 10 kilometer resolution using Gaussian process stacked generalization, an ensemble approach of multiple models. This approach has been shown to increase prediction accuracy for disease mapping compared with other methods such as Gaussian process regression[38]. This mapping procedure comprised two steps (Fig. 5b, c). In the first step, we trained three 'child models' to predict resistance prevalence based on a set of environmental and anthropogenic covariates, such as total antimicrobial use in 2013 and 2020, animal population density, and temperature (Supplementary Table 3; Supplementary Method). For each antimicrobial class, we also included the quantities (kg) used in 2020[18] disaggregated at 10 × 10 kilometer resolution as a covariate. This was calculated by disaggregating the total antimicrobial use per country proportionally to the distribution of animals' biomass in 2020[18]. Animals' biomass was calculated as the population correction units of food animals in 2020, using methods described in Van Boeckel et al. [39]. In the second step, the child model predictions were stacked using Gaussian process regression, fitted using the integrated nested Laplace approximations (INLA)[40] (Supplementary Methods). This second step allowed to simultaneously capture the influence of environmental and anthropogenic covariates, as well as the residual spatial correlation.

For each antimicrobial, we defined resistance hotspots as regions with resistance prevalence higher than the 95% percentile of all pixels on the map. We combined the drug-level resistance maps using summary metrics for the overall AMR level – N10, N25, or N50: the number of antimicrobials (out of 7) with resistance prevalence higher than 10%, 25%, or 50% in each pixel. For the summary AMR level across antimicrobial classes, resistance hotspots were defined as regions with N50 ≥ 3.

## Mapping priority antimicrobials for AMR surveillance

Priority antimicrobials for AMR surveillance were defined as antimicrobials that have the highest probability of their resistance prevalence exceeding a critical level (defined as 10%, 25%, or 50%) in the near future. Here, we assumed that prevalence of resistance will continue to increase in the future, based on temporal trends of AMR between 2000 and 2019. We developed an approach to predict priority antimicrobials at each 10 × 10 kilometer pixel, based on local risk factors as well as patterns of co-resistance in PPS. In the following, we explain the modeling process using 50% as the critical resistance level, while similar procedures were followed for the other cutoff values of resistance prevalence (10% or 25%). We illustrate the model formulation, with the following example of a pixel with N50 = 4 (Fig. 6).

Firstly, we binarized the resistance profile in 2015 for a given pixel (e.g. TET 70%, AMP 75%, SXT 60%, CHL 55%, CIP 40%, GEN 30%, and CTX 30%) by reclassifying the antimicrobials with resistance higher than 50% as 1, and the opposite as 0, such that the resistance profile for the 7 drugs considered in this analysis was: [1,1,1,1,0,0,0] (Fig. 6a). Secondly, for each of the three antimicrobials classified as 0 (e.g. CIP, GEN, CTX), we predicted whether their resistance prevalence will exceed 50% as a binary response variable (Fig. 6c), using covariates extracted from the collected surveys (Fig. 6d). The model considers future scenarios where only one additional antimicrobial will exceed 50% resistance (Fig. 6b). The model was constructed using least absolute shrinkage and selection operator (LASSO) applied to logistic regression. Using CIP as an example, its resistance prevalence exceeds 50% in resistance profile rp1, while it is absent in resistance profiles rp2 and rp3 (Fig. 6b, c). The covariates used to predict its presence and absence included two components. The first component considers patterns of co-resistance between antimicrobials, implying that probabilities of occurrence vary between resistance profiles. This variation is captured by using the proportion of surveys recording rp1 out of all surveys recording rp1, rp2 or rp3 as a covariate (Fig. 6d.i). Patterns of co-resistance also implies that the development of resistance of CIP is dependent on resistance of other antimicrobials. This dependence is captured by using the number of antimicrobials with resistance higher than 50% in the resistance profile in 2015 as a covariate (Fig. 6d.ii). The second component of covariates includes risk factors for predicting the development of resistance. This includes the percentage of CIP use (kg) out of all three antimicrobials at the location of the survey (Fig. 6d.iii), as well as a set of environmental and anthropogenic covariates associated with the locations of the surveys, such as total antimicrobial use in 2013 and 2020, temperature, and animal density (Fig. 6d.iv; Supplementary Table 3).

The above example was based on the current resistance profile rp0 (Fig. 6a). For CIP, there were in total 64 permutations of current

resistance profiles—all six antimicrobials apart from CIP could have resistance of 0 or 1. A complete model for CIP was trained by including all permutations in the procedure described in Fig. 6. This model was then applied to each pixel on the map where resistance to CIP has not yet exceeded 50%, to generate the probability that it will exceed 50% in the future. Similarly, the probabilities for the other antimicrobials were generated. Finally, at each pixel, we mapped the antimicrobial with the highest probability of its resistance prevalence exceeding 50% in the future.

The accuracy of the models for each antimicrobial was quantified by calculating the area under the receiver operating characteristic curve (AUC) using four-fold spatial cross validation[4]. The predictive capacity of the model came from two components of the covariates. The first component was based on co-resistance between drugs (Fig. 6d.i and Fig. 6d.ii). The second component was environmental and anthropogenic covariates associated with resistance to individual drugs (Fig. 6d.iii and Fig. 6d.iv). We quantified the relative contribution of these two covariate components to the model prediction accuracy, by calculating the drop in AUC following the withdrawal of each covariates compared with a full model including all covariates.

Furthermore, based on predictions of the priority antimicrobial for AMR surveillance at each $10 \times 10$ km pixel (Fig. 6), we estimated the time it takes for resistance prevalence of this antimicrobial to reach 50% in the future (Supplementary Fig. 16). Concretely, we extracted the current resistance prevalence estimated at each pixel, and calculated the time difference from the current resistance prevalence (Supplementary Fig. 16, time point a) until it reaches 50% (Supplementary Fig. 16, time point b), using the corresponding regression models fitted in section "Trends of AMR for each antimicrobial class".

### Uncertainty

The uncertainty of the mapped predictions of resistance prevalence (Fig. 5c) was calculated as the variance of the posterior predictive distribution for each map. The uncertainty of the mapped priority antimicrobials was calculated in two steps. Firstly, we generated 15 Monte Carlo simulations of imputed datasets of resistance prevalence, to incorporate the uncertainty introduced by imputation in the following analyses. Secondly, using the imputed datasets, we generated 15 maps of priority antimicrobials. We quantified its uncertainty by calculating—at each pixel—the proportion of maps that generated different predictions of antimicrobials as compared with the final map:

$$\text{Uncertainty} = \frac{N_{maps} \text{ with different predictions}}{m} \quad (1)$$

### Reporting summary

Further information on research design is available in the Nature Portfolio Reporting Summary linked to this article.

### Data availability

All data were extracted from literature reviews of point-prevalence surveys from PubMed (https://pubmed.ncbi.nlm.nih.gov), Scopus (https://www.scopus.com), ISI Web of Science (https://www.webofscience.com), and China National Knowledge Infrastructure (http://www.cnki.net). All data used for the analyses can be downloaded from the Figshare repository (https://doi.org/10.6084/m9.figshare.24231622), and can also be downloaded at resistancebank.org (https://resistancebank.org).

### Code availability

The codes used to generate the results are available at Zenodo (zenodo.org/record/8400343).

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

## Acknowledgements

C.Z. is supported by the Branco Weiss Fellowship. Y.W. and T.P.V.B. are supported by the Eccellenza Grant (no. PCEFP3_181248) from the Swiss National Science Foundation. R.M. is supported by the European Union's Horizon 2020 grant for MOOD (Monitoring Outbreaks for Disease Surveillance in a Data Science Context) (no. 874850).

## Author contributions

C.Z., Y.W., and R.M. collected data. C.Z. carried out the analysis. T.P.V.B. supervised the work. C.Z. wrote the initial manuscript. R.M. and T.P.V.B. revised the manuscript.

## Competing interests

The authors declare no competing interests.
