## [Peer Review File · Nature Communications]

Global surveillance of antimicrobial resistance in food animals using priority drugs mapsREVIEWER COMMENTS

Reviewer #1 (Remarks to the Author):

The noteworthy results are the maps showing the geographic distribution of antimicrobials with the highest probability of their resistance prevalence exceeding critical levels ie how many of the drugs that are currently in use already exceed 20, 35 and 50% resistance. This helps to concentrate on those antibiotics with high (but not extremely high) antibiotic resistance such as chloramphenicol to reduce the amount used (perhaps to have targets for use) in the affected countries.

The work will be of significance, it carries on from much of the research Thomas Van Boeckel has been doing on AMR in animals for many years.

The work does support the conclusions and claims, but in it is opaque in many areas, I would like to see more clarity around the numbers of animals (which breed) were included from each country and over how long (eg how many datapoints are included per year).

The only flaw I see in the methodology is lack of transparency for the collection of the data (years, countries, animals) to understand how sparse the data is.

The data would need to be available freely (is it freely available on Resistance Map?) with a link in the paper.

My Review follows.

Overview:

The researchers performed a systematic review of the literature extracting data from 1,015 point prevalence studies with prevalence of resistance data to seven antibiotics in two bacterial strains. They used geospatial mapping to understand the prevalence of resistance in the two important bacteria (*E. coli* and *Salmonella* species) and mapped these across low- and middle- income countries using additional covariates. The analysis was agnostic of the animal species being examined. They predicted hotspots in China, India, Brazil, Chile and part of central Asia and southeastern Africa.

Overall questions:

- Which animals were included needs to be described early on (together with the prevalence of resistance in the different animals), I see a breakdown in the Methods (data collection and imputation), but this would be useful to see visually as there is a big spread across the cattle, pigs, poultry etc – presumably this changes geographically as well. It would be useful to see this change mapped.
- How many surveys were included? The manuscript contains $n=1,015$ and in the supplement you mention 1,169 point prevalence surveys
- You predict 50% resistance into the future but give no dates. When you think this level will be reached/exceeded if there are no changes in practice?
- What methodology did the PPS's use? Was this harmonised or different across different countries?
- How many years were included in your systematic review? This needs to be described
- Were the PPS conducted on well animals (ie carriage), sick animals (disease) or those going to the abattoir? This needs more explanation in your methods section given there is no split by bacteria, the split by animals may be more important

Specific questions:

Figure 1:

- what does my_alpha mean as a key? Perhaps change this to number of surveys.
- Is this the number of global surveys (ie in some years up to 50 PPS surveys were conducted in LMICs)? If so, this number is very low.

Imputation of missing data on resistance prevalence for mapping priority antimicrobials

The narrative does not reflect the high imputation numbers eg you needed to impute 57% of the data for cefotaxime ($667/1169 = 57\%$), although less for the other bacteria, the imputation is still high for sulfamethoxazole-trimethoprim ($325/1169 = 27.8\%$), chloramphenicol ($291/1169 = 29\%$) etc in 1,169 point-prevalence surveys.

Supplementary figures 2&3:

- 1) box H: the colours don't seem to match on the map and the scale
- 2) how do you define MDR?

Supplementary figure 6 – this figure needs more explanation, the key suggests there won't be any prevalence over 50% for whichever bacteria (unclear) and whichever antibiotic (also unclear)

Supplementary figure 11:

- Include information on the numbers excluded (how many reviews/other non-PPS studies/strain surveys etc)
- What were the inclusion and exclusion dates for your systematic review?

Lines 284-5: lacking a reference for Murray et al.

Reviewer #2 (Remarks to the Author):

The manuscript "Antimicrobial resistance in food animals: priority drugs maps to guide global surveillance" aims at summarizing through the use of maps the information on the prevalence of AMR in two major bacterial species used for AMR monitoring, *E. coli* and *Salmonella*, available through point prevalence surveys in low- and middle-income countries (LMIC). Furthermore, it uses the information to predict possible trends in the near future using statistical modeling.

Given the importance of the topic under study, and the paucity of information on the situation of AMR in bacteria in many LMIC, the objectives proposed are important. The authors replicate the methodology they have already used in the past for similar purposes, and thus no major flaws in their approaches are expected. However, there are a number of things that should be further clarified in order to highlight what of new is this study bringing to the table, as well as to make it work as a stand alone article that does not require consulting other references in order to understand the way results were obtained. It seems to this reviewer that the main difference is the specific focus on *Salmonella* and *E. coli* and the imputation of missing data plus (perhaps?) the addition of another year to the dataset analyzed in Van Boeckel and Pires et al., 2019. Furthermore, because some of the methodology is only succinctly explained (in part because it had been used before), there may be room for misinterpretations. It would be good to highlight a few specifics in order to help the reader to make the most of this new study.

Comments:

- Clarification on data used: PPS estimates were extracted based on a literature review including surveys with data from the 2000-2019 period. Two references that would provide additional information in terms of the specifics of the search strings are included in the text (Van Boeckel and Pires et al., 2019, and Zhao et al., 2021) but the authors do not clearly state if in fact a full search was conducted specifically for this study or they just added the surveys from (late) 2019 to the data already collected for the first paper (that included studies published between 2000 and March 2019). Therefore, please indicate what the date of the search (used in this study) was, and also how does the database analyzed here overlaps (or not) with the one used in Van Boeckel and Pires et al., 2019, which included 901 surveys from several bacteria compared with 1,015 here from just *E. coli* and *Salmonella* (e.g., "adding the 2019 surveys resulted in XXX additional surveys, for a total of 380 in cattle, 292 in pigs, 540 in poultry, 84 in sheep and 2 in horse" or similar).
- Characteristics of dataset: because (apparently) there are some differences between the data analyzed here and the one included in previous studies, it would be helpful to add some information regarding methodologies used in the data analyzed here (e.g., percentage of reference complying with guidelines, etc.), similar to what was provided in Van Boeckel and Pires et al, since all I could find here was the number of studies per host species.
- Furthermore, according to the main manuscript, imputation of missing data was performed in 745 PPS in which at least 4 out of the 7 selected antimicrobials had been included, while in the supplementary methods (line 34) 1,169 surveys are mentioned. Similarly, according to the main text (lines 364-365) 9,145 is the total number of prevalence estimates used, of which 21% were imputed, while the supplementary methods (line 36) mention 8,855 prevalence estimates (of

which again the 21% were imputed). Please clarify and correct if needed.

- Trends of AMR: reading the methodology (and supplementary) it is not obvious to this reviewer if the authors calculated separate trends depending on the region (something that would be reasonable to at least check, in order to see if trends are in fact comparable across regions). While getting an overall estimate like the one shown in figure 1 is of course also useful, given that the objective is to capture the spatial variability in AMR in LMIC, some discussion (and/or additional analysis?) on regional differences depending on the trend would be useful.

- According to figure 3B there is considerable uncertainty in several regions (e.g., west part of Brazil) in some regions (>60-70% perhaps, though it is difficult to say from the color coding used in this figure). However, there is no mention in the text to Figure 3B (only to 3A) and no summary of the degree of uncertainty in the predictions. It would be good to add some information regarding e.g. number of pixels with uncertainty levels above a certain level (30-50%?) in the predictions.

Minor points:

- Line 35: given that non-therapeutical use of antimicrobials is (slowly, but still) decreasing, please add something like "in certain regions of the world" or similar at the end of the sentence

- Lines 43-45: the reference here (9) is based on data from Canada, so perhaps you could have "North America" or "US and Canada" instead of just US at the beginning of the sentence in line 43. Furthermore, ceftiofur is a drug and not an antimicrobial class, so consider having "e.g. cephalosporins such as ceftiofur" or similar given that the sentence seems to be based on "limitation of certain classes of antimicrobials".

- Lines 66-69: the authors speculate on the usefulness of the identification of antimicrobials for which resistance will go above a certain level (e.g., >50%) in a given region in the future as a tool to "inform veterinarians on possible treatment options". While I agree in principle, in order for this to happen in reality it would be necessary to measure resistance considering clinical breakpoints adapted to animals, since while a treatment is still effective in the field veterinarians and farmers are likely to still use it (even if the resistance according to CLSI breakpoints is above 50 or 80%). It would be good to add somewhere in the discussion the need for considering animal breakpoints if animal health is going to be a driver of veterinary AMU. Furthermore, the two bacteria selected here (*E. coli* and *Salmonella*) can be often found in apparently healthy animals and presumably a large proportion of the references used here were based on sampling of non-diseased animals. Therefore, some discussion on the need to consider other (not necessarily zoonotic but still relevant!) animal pathogens in this kind of exercises (if the point is to help "inform veterinarians on possible treatment options" would be useful to advance towards a real one health approach to the issue of AMR (and AMU). I agree that having a high resistance to a given antimicrobial in e.g. commensal *E. coli* is certainly signaling a problem, but this may have only limited implications in terms of treatment if 1) diseases are mediated by other pathogens and 2) treatment can be still successful as currently performed.

Reviewer #3 (Remarks to the Author):

Overall, this is a welcome addition to the work on AMR in food animals in LMICs as the previously reported P50 measure is not as actionable as presenting estimates for individual antibiotics or antibiotic classes. There's a lot of value in describing the current prevalence of resistance and identifying hotspots where intensified efforts at slowing the progression of resistance are needed. The estimates rely on robust data from point prevalence surveys with more limited geographic coverage in parts of South America and central-sub Saharan Africa. There are appropriate adjustments for changes in breakpoints across time, methodology, and standard used.

There are a number of questions and concerns that are mostly driven by the estimates related to forecasting:

The forecasting methods are unclear. The independent variables for the models end at 2019 or 2020 so what is being used to inform the prediction? What, precisely, is the dependent variable and what's informing the time to resistance exceeding 50% estimate? Does the point estimate of prevalence of resistance for a given pixel inform this model at all or is it only the dichotomized

resistance above or below 50%?

The result of the forecasting is surprising, the map in figure 3A suggests that tetracyclines are the antimicrobial with the highest probability of >50% resistance for most pixels but the time until resistance exceeds 50% is 5.1 years for tetracyclines compared to 1.7 years for penicillins and 4.1 years for cephalosporins.

In table S2 why are aminoglycosides NA?

The trends of AMR by antimicrobial class using linear regression for tetracycline and ampicillin is a little concerning and is conceptually a bit tough to understand why these two antibiotics in particular would violate linearity using the logit of the dependent variable; for example, the ampicillin is being modeled with a linear regression when the cefotaxime is logistic regression, this is surprising since they're broadly part of the same antibiotic class - i.e. both beta lactam antibiotics and mechanism of resistance often leads to overlapping resistance and probably deserves some further comment or exploration as to why this is plausible. It's also noteworthy that tetracycline and ampicillin are the two antibiotics most commonly identified by the authors as priority antibiotics and also happen to utilize a model distinct from the other 5 antimicrobials.

I would be interested to know how the candidate covariates were selected – e.g. percentage of tree cover.

Can the authors comment on the coefficients for the proportion_PPS covariate being positive for some antimicrobials and negative for others? I assume the proportion_PPS covariate is what's described in figure 5 (d)[i] if so I would seek clarification on whether a profile where CIP, GEN, and CTX are all >50% resistance would be included in rp1 (ie in the numerator) or excluded altogether. I assume it's the former but it's not clear. Is proportion_PPS location specific? If a location in India and Brazil have the same current resistance profile (rp) would they also have the same proportion_PPS? The labeling in figure 5 might need to be reviewed, particularly the second column 'future resistance profile,' it's not clear that a temporal relationship is being enforced in the production of the proportion_PPS covariate.

Figure 2 should include a year, I would also advocate for including all 7 antimicrobials in the figure.

The work this seems to build on produced estimates by animal type, this work moved away from that and didn't seem to include every animal type as covariates according to table S3. I'd welcome the author's comments on that shift since it seemed like they identified resistance rising in chickens faster than other food animals.

I appreciate the representation of uncertainty but it would also be helpful to know the number of isolates informing the estimates so that readers can better appreciate the limitations or strengths of the data.

The discussion focuses a lot on priorities for surveillance and not enough on measures to reduce antimicrobial use in food animals.

Response to reviewers #NCOMMS-23-2661

The comments from the reviewers are in **black**, the responses to reviewers and modifications to the original manuscript are in **blue**, quotes to the original manuscript are provided in **grey**.

Reviewer #1 (Remarks to the Author):

The noteworthy results are the maps showing the geographic distribution of antimicrobials with the highest probability of their resistance prevalence exceeding critical levels ie how many of the drugs that are currently in use already exceed 20, 35 and 50% resistance. This helps to concentrate on those antibiotics with high (but not extremely high) antibiotic resistance such as chloramphenicol to reduce the amount used (perhaps to have targets for use) in the affected countries.

The work will be of significance, it carries on from much of the research Thomas Van Boeckel has been doing on AMR in animals for many years.

The work does support the conclusions and claims, but in it is opaque in many areas, I would like to see more clarity around the numbers of animals (which breed) were included from each country and over how long (eg how many datapoints are included per year).

The only flaw I see in the methodology is lack of transparency for the collection of the data (years, countries, animals) to understand how sparse the data is.

The data would need to be available freely (is it freely available on Resistance Map?) with a link in the paper.

We thank the reviewer for the assessment and suggestions. Our revision focuses on providing clarity in the collection of the data, and a breakdown of the datapoints per animal species, year, and country is now provided as supplementary material in Supplementary Table 9 and 10. The reviewer is right that all data extracted from point-prevalence surveys for the analyses will be freely available for download as the Supplementary Dataset for the manuscript, as well as from <https://resistancebank.org> upon publication of the manuscript.

Year	Chicken	Pigs	Cattle
2000	1	0	1
2001	0	0	1
2002	2	2	2
2003	10	4	6
2004	0	0	5
2005	2	1	2
2006	6	3	6
2007	12	4	11
2008	11	12	14
2009	12	8	11
2010	31	14	18
2011	17	16	22
2012	50	12	19
2013	33	16	34
2014	63	38	46
2015	56	32	42
2016	85	31	33
2017	70	39	60
2018	50	31	41
2019	59	40	35

Supplementary Table 9. Number of point-prevalence surveys conducted on chicken, pigs, and cattle in each year.

Country	Chicken	Pigs	Cattle
AGO	1	1	1
BFA	2	1	2
BWA	1	1	3
CMR	1	0	0
DZA	8	0	2
EGY	17	0	12
ETH	11	2	26
GAB	1	0	0
GHA	3	0	2
GMB	1	0	0
KEN	5	2	2
MAR	6	0	4
NAM	0	0	1
NGA	16	4	10
SDN	0	0	2
SEN	1	0	1
TCD	1	0	0
TUN	13	0	8
TZA	3	1	7
UGA	4	2	3
ZAF	5	6	10
ZMB	1	0	2
ZWE	2	0	1
RWA	1	0	0
IND	84	20	93
BGD	36	0	11
NPL	8	1	1
BTN	1	1	0
PAK	10	0	4
IRN	33	0	21
IRQ	6	0	2
ISR	2	0	2
LBN	3	1	4
QAT	1	0	0
OMN	1	0	0
JOR	0	0	3
SAU	2	0	1
TUR	11	0	8
ARG	5	8	5
BOL	1	0	0
BRA	28	13	27
COL	1	1	4
ECU	6	0	0
PER	2	1	1
VEN	1	1	2
CHL	0	0	3
CRI	1	0	0
MEX	6	5	15
GRD	2	0	0
LCA	0	0	0
JAM	1	0	0
CHN	169	186	75
IDN	3	0	5
KHM	2	3	0
LAO	0	2	2
MYS	6	2	4
MMR	1	0	0
PHL	1	3	0
SGP	1	0	0
THA	14	26	10
VNM	14	9	7
LKA	1	0	0
KWT	1	0	0
PRY	1	0	0

Supplementary Table 10. Number of point-prevalence surveys conducted on chicken, pigs, and cattle in each country.

My Review follows.

Overview:

The researchers performed a systematic review of the literature extracting data from 1,015 point prevalence studies with prevalence of resistance data to seven antibiotics in two bacterial strains. They used geospatial mapping to understand the prevalence of resistance in the two important bacteria (*E. coli* and *Salmonella* species) and mapped these across low- and middle- income countries using additional covariates. The analysis

was agnostic of the animal species being examined. They predicted hotspots in China, India, Brazil, Chile and part of central Asia and southeastern Africa.

Overall questions:

- Which animals were included needs to be described early on (together with the prevalence of resistance in the different animals), I see a breakdown in the Methods (data collection and imputation), but this would be useful to see visually as there is a big spread across the cattle, pigs, poultry etc – presumably this changes geographically as well. It would be useful to see this change mapped.

We thank the reviewer for suggesting to break down the trends of resistance based on the animal species tested. For each animal species, we plotted the temporal trends of resistance in Supplementary Figures 1 - 3, and included the following description in lines 106 – 110 in the manuscript:

“Prevalence of resistance was investigated in poultry in 52% of PPS, in cattle in 38% of PPS, and in pigs in 28% of PPS. Prevalence of resistance increased significantly for AMP, CHL, CIP, and CTX for poultry, and for AMP, SXT, CHL, CIP, GEN, and CTX for pigs (Supplementary Figures 1 to 2). However, temporal trends of resistance were not significant for any antimicrobial classes for cattle (Supplementary Figures 3).”

Supplementary Figure 1. **Chicken:** temporal trends of the prevalence of resistance, for ampicillin (AMP), chloramphenicol (CHL), ciprofloxacin (CIP), cefotaxime (CTX), gentamicin (GEN), sulfamethoxazole-trimethoprim (SXT), and tetracycline (TET). Solid lines represent significant temporal trends ($p < 0.05$), and dashed lines represent nonsignificant trends. Transparency levels of the red colors were proportional to the number of surveys published each year. Temporal trends were significant (p value < 0.05) for AMP, CHL, CIP, and CTX.

Supplementary Figure 2. **Pigs:** Temporal trends of the prevalence of resistance, for ampicillin (AMP), chloramphenicol (CHL), ciprofloxacin (CIP), cefotaxime (CTX), gentamicin (GEN), sulfamethoxazole-trimethoprim (SXT), and tetracycline (TET). Solid lines represent significant temporal trends ($p < 0.05$), and dashed lines represent nonsignificant trends. Transparency levels of the red colors were proportional to the number of surveys published each year. Temporal trends were significant (p value < 0.05) for all antimicrobials except TET.

Supplementary Figure 3. **Cattle**: temporal trends of the prevalence of resistance in cattle, for ampicillin (AMP), chloramphenicol (CHL), ciprofloxacin (CIP), cefotaxime (CTX), gentamicin (GEN), sulfamethoxazole-trimethoprim (SXT), and tetracycline (TET). Solid lines represent significant temporal trends ($p < 0.05$), and dashed lines represent nonsignificant trends. Transparency levels of the red colors were proportional to the number of surveys published each year. Temporal trends were not significant ($p > 0.05$) for all antimicrobials.

The reviewer is correct that the animal species tested in the surveys differed geographically. We mapped the geographic locations of surveys conducted for each animal species in Supplementary Figure 13 to provide clarity on that point.

Supplementary Figure 13. Geographic locations of point-prevalence surveys reporting resistance prevalence of *E. coli* and *Salmonella* isolated from poultry (A), pigs (B), and cattle (C). Sizes of the circle were in proportion to the log₁₀ transformed sample sizes of each survey. Colors of the circles represented the prevalence of resistance reported in each survey.

• How many surveys were included? The manuscript contains $n=1,015$ and in the supplement you mention 1,169 point prevalence surveys

We thank the reviewer for pointing out the error here. There were 1,015 publications included in the analysis. We mistakenly reported a higher number in the SI, because some publications tested resistance to multiple animal species, and reported resistance profiles for each animal species separately. The 1,169 here was referring to the total number of resistance profiles used for imputation – instead of the number of publications. This has now been corrected.

Additionally, since the initial submission (June 21st), we have completed an additional literature review of 73 surveys published in 2019. These surveys have been added to the present analysis and all outputs revised accordingly. Therefore, the present analysis now includes in total 1,088 surveys.

We added the following sentence in the SI to clarify the difference between the number of PPS and the number of resistance profiles: “We conducted imputation on a subset of 806 PPS that reported at least 4 out of the 7 drugs. These PPS contained 1,411 resistance profiles – some PPS reported resistance profiles for multiple animal species or for multiple sample types.”

- You predict 50% resistance into the future but give no dates. When you think this level will be reached/exceeded if there are no changes in practice?

Thank you for pointing out that the predicted time ranges for reaching 50% resistance were originally provided in relative terms rather than absolute terms (a specific year): “Across locations, the average estimated time for resistance prevalence to exceed 50%, weighted by animals’ biomass (Methods), ranged from 1.7 years for AMP to 12.4 years for CIP (Supplementary Table 2; Supplementary Figure 6).” Making projections in absolute terms is by nature challenging because the datapoints used to train the predictive model were collected over multiple years (between 2000 and 2019). However, we do acknowledge the relevance of the referee’s remark in providing a baseline year from which to apply these predictions, and have used the median year of the training data, 2015. All projections for reaching 50% resistance have been adjusted accordingly, and are now reflected in the manuscript (lines 299 – 312):

“We estimated the time it may take for resistance prevalence of the predicted priority antimicrobials to exceed a critical level. For locations where tetracyclines, penicillins or cephalosporins were the predicted priority antimicrobials, the average time for resistance to reach 50% across locations was below 7 years. Given that the median year of publication of the PPS was 2015, this implies that resistance may have already exceeded 50% now at these locations. For locations where amphenicols or quinolones were the predicted priority antimicrobials, their prevalence of resistance was estimated to exceed 50% in 2026 and 2027 on average across locations. However, the temporal trends of AMR used for estimating the time was based on data across low- and middle-income countries, and may differ depending on the geographic region. Estimating separate trends for each region is challenged by the limited amount of data in America and Africa countries, with large uncertainty associated with the estimated coefficients (Supplementary Table 2). Future work may be able to make region-specific projections of AMR as the number of PPS published each year steadily increases and more data becomes available.”

- What methodology did the PPS’s use? Was this harmonised or different across different countries?

The methodology varies between surveys, with disk diffusion being used in the majority of PPS (79%). In section “Literature review and data extraction” in Supplementary Information, we added a description on the methodologies used in PPS:

“Antimicrobial susceptibility testing in the PPS was conducted using either diffusion methods or dilution methods. The majority of PPS used diffusion methods, including disk diffusion (79%) and E-test (0.2%). The rest of PPS used dilution methods, including broth dilution (14%), agar dilution (5%), and automated devices such as VITEK2 (2%). Among the PPS, there was no systematic difference in the measurements between these two families of methods (Van Boeckel & Pires *et al.* 2019). In each PPS, antimicrobial susceptibility testing results are compared with breakpoints to determine resistance, which are provided by laboratory guidelines and revised annually. Only 18% of records reported the breakpoints used. However, the majority (93%) of PPS mentioned the name of laboratory guidelines used, and 66% among these also mentioned the year of the guideline. The guidelines mentioned by the PPS included guidelines published by the Clinical & Laboratory Standards Institute (96%), the European Committee on Antimicrobial Susceptibility Testing (3%), and the French Society of Microbiology (1%). We adjusted for variations of breakpoints used between surveys, using a method developed by Van Boeckel and Pires *et al.* 2019 in section “Harmonization of Antimicrobial Resistance Rates” in the Supplementary Material of the reference publication. The adjustment resulted in 635 (2%) resistance prevalence being revised.”

- How many years were included in your systematic review? This needs to be described

In lines 383 - 385 in the Methods section, we included the number of years in the literature review:

“We extracted 1,088 point prevalence surveys on AMR of *E. coli* and *Salmonella* in healthy food animals across low- and middle-income countries (LMICs) across two decades between 2000 and 2019”

We included this information in section “Literature review and data extraction” in the Supplementary Information as well, with details on when the literature searches were conducted and for which years:

“We searched for point-prevalence surveys (PPS) published between 2000 and 2019 reporting antimicrobial resistance in healthy food animals in low- and middle-income countries, focusing on *Escherichia coli* and

nontyphoidal *Salmonella* spp.. The literature search was conducted in three rounds from four databases - PubMed, Scopus, ISI Web of Science, and China National Knowledge Infrastructure. The first round was conducted on 28.03.2019 from the first three aforementioned databases, and extracted data from all papers published between January 2000 and December 2018. The extracted data and details of literature review were published in Van Boeckel and Pires *et al.* 2019. The second round of literature search was conducted on 11.03.2020 from all four databases, and included surveys published between January 2000 and December 2019 exclusively for China. The extracted data and details of literature review were published in Zhao *et al.* 2020. The third round of literature search was conducted on 12.01.2022 from the first three aforementioned databases, and included all papers published between January 2019 and December 2019 in low- and middle-income countries apart from China. The search queries used for the third round of literature review was the same as in Van Boeckel and Pires *et al.* 2019.”

- Were the PPS conducted on well animals (ie carriage), sick animals (disease) or those going to the abattoir? This needs more explanation in your methods section given there is no split by bacteria, the split by animals may be more important

Our analysis included PPS conducted on healthy animals. In line 383 in the Methods section, we changed “food animals” to “healthy food animals” for clarity. In lines 390 to 394, we included the following description on sample types:

“The animal samples used to determine resistance prevalence were taken from their meat (34% of total resistance prevalence), swabs from living animals on farm or in wet markets (32%), food products such as milk and eggs (16%), swabs from slaughtered animals (9%), and fecal samples on farm (7%).”

Specific questions:

Figure 1:

- what does my_alpha mean as a key? Perhaps change this to number of surveys.

We thank the reviewer for pointing this out. We changed “my_alpha” to “Number of Surveys”.

- Is this the number of global surveys (ie in some years up to 50 PPS surveys were conducted in LMICs)? If so, this number is very low.

The reviewer is right that the number here represents the number of global surveys. In the earlier years, the number of surveys available are indeed limited. Between 2000 and 2005, there were only 31 surveys available. However, this number is growing, and 735 surveys were available between 2014 and 2019. We added the following in the legend for Figure 1 to clarify that the number represents the total number of surveys in all LMICs:

“Transparency levels of the red colors were proportional to the number of surveys published each year across low- and middle-income countries.”

Imputation of missing data on resistance prevalence for mapping priority antimicrobials

The narrative does not reflect the high imputation numbers eg you needed to impute 57% of the data for cefotaxime (667/1169 = 57%), although less for the other bacteria, the imputation is still high for sulfamethoxazole-trimethoprim (325/1169 = 27.8%), chloramphenicol (291/1169 = 29%) etc in 1,169 point-prevalence surveys.

The reviewer is right that the numbers of imputation are high especially for cefotaxime. We added the percentages beside the corresponding numbers in the Supplementary Information (Note that the numbers of resistance prevalence are now higher compared with last round of review, because we included more surveys into the analysis pipeline):

“The number of imputed resistance prevalence was 720 for cefotaxime (51%), 375 (27%) for sulfamethoxazole-trimethoprim, 306 (22%) for chloramphenicol, 202 (14%) for ampicillin, 196 (14%) for tetracycline, 195 (14%) for ciprofloxacin, and 123 (9%) for gentamicin.”

We also added the high numbers of imputation as a limitation in lines 323 - 328:

“Firstly, predictive maps, as well as the imputation of missing resistance prevalence for modelling priority antimicrobials introduces uncertainty. The number of imputations was highest for cefotaxime – its resistance prevalence was missing in half (51%) of the surveys. However, the uncertainty of the missing values was captured by the high standard deviation (24%) of the multiple imputed values for cefotaxime.”

Supplementary figures 2&3:

1) box H: the colours don't seem to match on the map and the scale

Thank you for pointing out the problem with color scales. We originally marked all areas with the number of antimicrobials with resistance higher than 50% (N_{50}) ≥ 3 in cyan color, and this caused a lack of differentiation in the color of the map. We now removed the cyan mark for better differentiation.

Additionally, note that supplementary Figure 2&3 are now shown in Figure 2 in the main manuscript.

2) how do you define MDR?

Due to the different definitions of MDR under different contexts – e.g. often defined as resistance to at least three antimicrobial classes but sometimes referring to resistance to a certain set of antimicrobials (see for example <https://www.microbiologyresearch.org/content/journal/jmm/10.1099/jmm.0.46747-0>), and in order to avoid confusion raised by the phrase, we revised "multi-drug resistance" to "overall resistance level across antimicrobials measured using N_{50} " in the legend of Figure 2. N_{50} can have values from 0 to 7.

Supplementary figure 6 – this figure needs more explanation, the key suggests there won't be any prevalence over 50% for whichever bacteria (unclear) and whichever antibiotic (also unclear)

We apologize for the lack of clarity, and have revised Supplementary Figure 10 (previously numbered Supplementary Figure 6) to include an additional panel to allow both maps to be read side-by-side. The values shown on each location in panel A indicates the number of years it will take for resistance of the corresponding antimicrobial shown in panel B to exceed 50%. For better clarity, we revised lines 165 – 169 in the manuscript from

"Across locations, the average estimated time for resistance prevalence to exceed 50%, weighted by animals' biomass (Methods), ranged from 1.7 years for AMP to 12.4 years for CIP (Supplementary Table 2; Supplementary Figure 10)."

to

"We estimated the time for resistance prevalence to exceed 50%, for the predicted priority antimicrobial in each 10 x 10 km pixel (Supplementary Figure 10). Across locations where AMP was the predicted priority antimicrobial (Supplementary Figure 10B), the average time weighted by animals' biomass was 1.7 years, while for CIP the average time was 12.4 years (see Supplementary Table 4 for the average estimated time for each antimicrobial class)."

Additionally, we reversed the color scale for Supplementary Figure 10A, such that the darker color represents locations where resistance is predicted to reach 50% sooner.

Supplementary Figure 10. Estimated time (years) that it takes for the prevalence of resistance to exceed 50% (A), for the predicted antimicrobial with the highest probability of its resistance prevalence exceeding 50% in the future (B).

Supplementary figure 11:

- Include information on the numbers excluded (how many reviews/other non-PPS studies/strain surveys etc)

We added the numbers excluded in each step, for the three rounds of literature review individually. A few of the numbers for the first round of literature review that was conducted in 2019 were unfortunately unable to be recovered, and these were written as NA:

Exclusion Criteria	Literature Review Round #1	Literature Review Round #2	Literature Review Round #3
	n _{hits} = 32,030	n _{hits} = 8,481	n _{hits} = 3,814
Reviews, meta-analysis, and other non-PPS studies	- 30,038	- 7,401	- 3,263
	n _{screened} = 1,992	n _{screened} = 1,080	n _{screened} = 551
Strain Surveys	NA	-115	-33
Diseased Animals	NA	-164	-21
Mixed Samples	NA	-97	-92
No Geographic Data	NA	-13	-54
Others	NA	-370	-238
	n _{PPS} = 926	n _{PPS} = 321	n _{PPS} = 113
Small Species Sample Size	-25	-6	-4
Drug-Pathogen Combinations Not Considered	-156	-45	-36

Supplementary Table 1. Extraction of point prevalence surveys (PPS) of antimicrobial resistance in *E. coli* and *Salmonella*, and exclusion criteria.

- What were the inclusion and exclusion dates for your systematic review?

The inclusion and exclusion of the publications were conducted over periods that lasted over months for each round of literature review, and were therefore not associated with concrete dates. We added the dates of the literature search – corresponding to when the inclusion and exclusion procedure starts – in the Supplementary Information:

“The first round was conducted on 28.03.2019

The second round of literature search was conducted on 11.03.2020

The third round of literature search was conducted on 12.01.2022.”

Lines 284-5: lacking a reference for Murray et al.

Thank you, we added the reference now.

Reviewer #2 (Remarks to the Author):

The manuscript “Antimicrobial resistance in food animals: priority drugs maps to guide global surveillance” aims at summarizing through the use of maps the information on the prevalence of AMR in two major bacterial species used for AMR monitoring, *E. coli* and *Salmonella*, available through point prevalence surveys in low- and middle-income countries (LMIC). Furthermore, it uses the information to predict possible trends in the near future using statistical modeling.

Given the importance of the topic under study, and the paucity of information on the situation of AMR in bacteria in many LMIC, the objectives proposed are important. The authors replicate the methodology they have already used in the past for similar purposes, and thus no major flaws in their approaches are expected. However, there are a number of things that should be further clarified in order to highlight what of new is this study bringing to the table, as well as to make it work as a stand alone article that does not require consulting other references in order to understand the way results were obtained. It seems to this reviewer that the main difference is the specific focus on *Salmonella* and *E. coli* and the imputation of missing data plus (perhaps?) the addition of another year to the dataset analyzed in Van Boeckel and Pires et al., 2019. Furthermore, because some of the methodology is only succinctly explained (in part because it had been used before), there may be room for misinterpretations. It would be good to highlight a few specifics in order to help the reader to make the most of this new study.

We thank the reviewer for the feedback. We revised the manuscript to better highlight the original contributions of this study as compared with Van Boeckel & Pires et al. 2019. First, we broke down trends of AMR by antimicrobial classes as compared with the previous paper that used a summary metric to map hotspots of AMR that may be of limited use for targeting interventions for concrete drug classes. We added more information on drug-specific trends in section “Trends of AMR” in the manuscript, and in Supplementary Figures 1 – 6. Our estimated trends were also fitted with more data as detailed in the new section “Literature review and data extraction” in the Supplementary Information. Second, we attempted to provide information on which drug classes to prioritize for intervention not only from a geographic perspective but also from a temporal perspective,

by predicting the drug classes that will most likely exceed certain resistance levels in the future. Third, from a methodological perspective, we leveraged both co-resistance patterns and local risk factors to make inference on the drug level which – to the best of our knowledge – have not been attempted on large spatial scales. We revised the Method section to better illustrate the methodology.

Comments:

- Clarification on data used: PPS estimates were extracted based on a literature review including surveys with data from the 2000-2019 period. Two references that would provide additional information in terms of the specifics of the search strings are included in the text (Van Boeckel and Pires et al., 2019, and Zhao et al., 2021) but the authors do not clearly state if in fact a full search was conducted specifically for this study or they just added the surveys from (late) 2019 to the data already collected for the first paper (that included studies published between 2000 and March 2019). Therefore, please indicate what the date of the search (used in this study) was, and also how does the database analyzed here overlaps (or not) with the one used in Van Boeckel and Pires et al., 2019, which included 901 surveys from several bacteria compared with 1,015 here from just *E. coli* and *Salmonella* (e.g., “adding the 2019 surveys resulted in XXX additional surveys, for a total of 380 in cattle, 292 in pigs, 540 in poultry, 84 in sheep and 2 in horse” or similar).

We thank the reviewer for pointing out the lack of clarity in our description of the literature review. The database used in this analysis is bigger than that used in Van Boeckel and Pires et al. 2019, as we included data from the whole year of 2019. The literature review was conducted in three rounds: the first round was associated with the publication of Van Boeckel and Pires et al. 2019, where surveys until 2018 were collected from three international search engines. The second round was associated with Zhao et al. 2021, where surveys in China until 2019 were collected from a Chinese-language search engine, as well as the aforementioned three international search engines. The third round was just completed, and included surveys from countries other than China until 2019. At the time of last submission, we were still in the process of finishing the third round of literature review. Now we have finalized this part, and added 73 surveys in our analysis and updated our estimates of AMR trends.

We included a section “Literature review and data extraction” in the Supplementary Information with details on the literature review process:

“Literature review and data extraction

We searched for point-prevalence surveys (PPS) published between 2000 and 2019 reporting antimicrobial resistance in healthy food animals in low- and middle-income countries, focusing on *Escherichia coli* and nontyphoidal *Salmonella* spp.. The literature search was conducted in three rounds from four databases - PubMed, Scopus, ISI Web of Science, and China National Knowledge Infrastructure. The first round was conducted on 28.03.2019 from the first three aforementioned databases, and extracted data from all papers published between January 2000 and December 2018. The extracted data and details of literature review were published in Van Boeckel and Pires *et al.* 2019. The second round of literature search was conducted on 11.03.2020 from all four databases, and included surveys published between January 2000 and December 2019 exclusively for China. The extracted data and details of literature review were published in Zhao *et al.* 2020. The third round of literature search was conducted on 12.01.2022 from the first three aforementioned databases, and included all papers published between January 2019 and December 2019 in low- and middle-income countries apart from China. The search queries used for the third round of literature review was the same as in Van Boeckel and Pires *et al.* 2019.

All three rounds of literature review were conducted with the following procedure (Supplementary Table 3). First, we screened in total 44,325 titles and abstracts, and excluded 40,702 non-PPS publications. We read 3,623 manuscripts in full, and excluded strain surveys, surveys on diseased animals, surveys conducted on a mixture of animal species, surveys without subnational geographic information, and other non-PPS surveys. After the exclusion, there were 1,360 PPS suitable for AMR mapping purposes. We further excluded animal species with small sample sizes such as camel and buffalo, and excluded drug-pathogen combinations not considered in this analysis such as *Campylobacter* and erythromycin. After the exclusion, 1,088 PPS that reported resistance prevalence in *Escherichia coli* and nontyphoidal *Salmonella* spp to 7 antimicrobials (listed in Methods section) were retained for the analyses. All data used in the current analyses are available in the supplementary file, and can also be downloaded at <https://resistancebank.org>.”

We also added the following in the Methods section that referred to the above explanation in the SI:

“These surveys were collected through three rounds of literature review of four databases (PubMed, Scopus, ISI Web of Science, and China National Knowledge Infrastructure). The process of data extraction is explained in detail in the Supplementary Information section “Literature review and data extraction”.”

- Characteristics of dataset: because (apparently) there are some differences between the data analyzed here and the one included in previous studies, it would be helpful to add some information regarding methodologies used in the data analyzed here (e.g., percentage of reference complying with guidelines, etc.), similar to what

was provided in Van Boeckel and Pires et al, since all I could find here was the number of studies per host species.

We included information regarding methodologies used in the PPS in the Supplementary Information section “Literature review and data extraction”:

“Antimicrobial susceptibility testing in the PPS was conducted using either diffusion methods or dilution methods. The majority of PPS used diffusion methods, including disk diffusion (79%) and E-test (0.2%). The rest of PPS used dilution methods, including broth dilution (14%), agar dilution (5%), E-test (0.2%), and automated devices such as VITEK2 (2%). For both families of methods, antimicrobial susceptibility testing results are compared with breakpoints to determine resistance. Only 18% of records reported the breakpoints used. However, the majority (93%) of PPS mentioned the name of laboratory guidelines used, and 66% among these also mentioned the year of the guideline. The guidelines mentioned by the PPS included guidelines published by the Clinical & Laboratory Standards Institute (96%), the European Committee on Antimicrobial Susceptibility Testing (3%), and the French Society of Microbiology (1%). We adjusted for variations of breakpoints used between surveys, using a method developed by Van Boeckel and Pires *et al* 2019¹. The adjustment resulted in 635 (2%) resistance prevalence being revised.”

- Furthermore, according to the main manuscript, imputation of missing data was performed in 745 PPS in which at least 4 out of the 7 selected antimicrobials had been included, while in the supplementary methods (line 34) 1,169 surveys are mentioned.

We thank the reviewer for pointing out the mistake here. There were 745 publications (now 806 publications as we included more PPS in the analysis) which reported at least 4 of the selected antimicrobials. We mistakenly reported a higher number in the SI, because some publications tested resistance to multiple animal species, and reported resistance profiles for each animal species separately. The 1,169 here (now 1,411) was referring to the total number of resistance profiles – instead of the number of publications. We revised the description in the SI for better clarity:

“We conducted imputation on a subset of 806 PPS that reported at least 4 out of the 7 drugs. These PPS contained 1,411 resistance profiles – some PPS reported resistance profiles for multiple animal species or for multiple sample types.”

Similarly, according to the main text (lines 364-365) 9,145 is the total number of prevalence estimates used, of which 21% were imputed, while the supplementary methods (line 36) mention 8,855 prevalence estimates (of which again the 21% were imputed). Please clarify and correct if needed.

We thank the reviewer for pointing out the mistake here. In both places, the number should have been 8,855. As we added more surveys in the analysis now, the number is revised to 9,877 (1,411 resistance profiles times 7 drugs in each profile) now. We have revised the number in both places.

- Trends of AMR: reading the methodology (and supplementary) it is not obvious to this reviewer if the authors calculated separate trends depending on the region (something that would be reasonable to at least check, in order to see if trends are in fact comparable across regions). While getting an overall estimate like the one shown in figure 1 is of course also useful, given that the objective is to capture the spatial variability in AMR in LMIC, some discussion (and/or additional analysis?) on regional differences depending on the trend would be useful.

We thank the reviewer for suggesting to estimate temporal trends for each region separately. We plotted resistance trends for LMICs in Asia, America, and Africa (Supplementary Figures 4 - 6). We summarized the coefficient associated with the temporal trends for the logistic regression in Supplementary Table 2.

Supplementary Figure 4. Asia: temporal trends of the prevalence of resistance for ampicillin (AMP), chloramphenicol (CHL), ciprofloxacin (CIP), cefotaxime (CTX), gentamicin (GEN), sulfamethoxazole-trimethoprim

(SXT), and tetracycline (TET). Solid lines represent significant temporal trends ($p < 0.05$), and dashed lines represent nonsignificant trends. Transparency levels of the red colors were proportional to the number of surveys published each year. Temporal trends were significant ($p < 0.05$) for AMP, CHL, CIP, and CTX.

Supplementary Figure 5. **Africa:** Temporal trends of the prevalence of resistance for ampicillin (AMP), chloramphenicol (CHL), ciprofloxacin (CIP), cefotaxime (CTX), gentamicin (GEN), sulfamethoxazole-trimethoprim (SXT), and tetracycline (TET). Solid lines represent significant temporal trends ($p < 0.05$), and dashed lines represent nonsignificant trends. Transparency levels of the red colors were proportional to the number of surveys published each year. Temporal trends were significant ($p < 0.05$) for TET and AMP.

Supplementary Figure 6. **America:** Temporal trends of the prevalence of resistance for ampicillin (AMP), chloramphenicol (CHL), ciprofloxacin (CIP), cefotaxime (CTX), gentamicin (GEN), sulfamethoxazole-trimethoprim (SXT), and tetracycline (TET). Solid lines represent significant temporal trends ($p < 0.05$), and dashed lines represent nonsignificant trends. Transparency levels of the red colors were proportional to the number of surveys published each year. Temporal trends were significant ($p < 0.05$) for CTX.

	All		Africa (n = 1,673)		Asia (n = 6,148)		America (1,023)	
	estimate	standard error	estimate	standard error	estimate	standard error	estimate	standard error
TET	0.027	0.014	0.07	0.033	0.016	0.018	-0.017	0.043
AMP	0.074	0.015	0.088	0.034	0.063	0.018	0.087	0.047
SXT	0.045	0.016	0.064	0.037	0.029	0.019	0.071	0.053
CHL	0.058	0.017	0.036	0.046	0.078	0.02	-0.024	0.059
CIP	0.051	0.018	0.081	0.054	0.046	0.02	0.076	0.063
GEN	0.037	0.016	0.035	0.053	0.031	0.018	0.118	0.061
CTX	0.155	0.03	0.109	0.059	0.148	0.037	0.319	0.107

Supplementary Table 2. Coefficients associated with logistic regressions on temporal trends of resistance prevalence for Africa, Asia and America. Significant ($p < 0.05$) coefficient values are shown in bold.

We added the following in the discussion section:

“However, the temporal trends of AMR used for estimating the time was based on data across low- and middle-income countries, and may differ depending on the geographic region. Estimating separate trends for each region is challenged by the limited amount of data in America and Africa countries, with large uncertainty associated with the estimated coefficients (Supplementary Table 2). Future work may be able to make region-specific projections of AMR as the number of PPS published each year steadily increases and more data becomes available.”

- According to figure 3B there is considerable uncertainty in several regions (e.g., west part of Brazil) in some regions (>60-70% perhaps, though it is difficult to say from the color coding used in in this figure). However, there is no mention in the text to Figure 3B (only to 3A) and no summary of the degree of uncertainty in the predictions. It would be good to add some information regarding e.g. number of pixels with uncertainty levels above a certain level (30-50%?) in the predictions.

Thank you for suggesting to include a summary of uncertainty levels. We added the following summary in lines 161 – 165: “The uncertainty associated with the predicted priority antimicrobials was on average 12% across all pixels (Figure 3B), and was high (> 40%) in parts of western Brazil, South Sudan and North Korea. The percentage of pixels with high uncertainty (> 40%) for each country was calculated in Supplementary Table 11 (see below).”

Supplementary Table 11. The percentage of 10 x 10 km pixels in each country where the uncertainty of the predicted priority antimicrobial is higher than 0.4.

Country ISO3	Percentage of pixels with uncertainty > 0.4	Country ISO3	Percentage of pixels with uncertainty > 0.4	Country ISO3	Percentage of pixels with uncertainty > 0.4
AFG	0.210069	GNB	0.164634	PAK	0.082604
AGO	0.030731	GNQ	0	PAN	0.109756
ARE	0.152968	GTM	0.103553	PER	0.141867
ARG	0.032319	GUY	0.120746	PHL	0.159052
ARM	0.148741	HND	0.034509	PRI	0.064103
AZE	0.059736	HTI	0.086806	PRK	0.766391
BDI	0.068729	IDN	0.142597	PRY	0.031902
BEN	0.081277	IND	0.071204	PSX	0.383721
BFA	0.059798	IRN	0.131975	QAT	0.163934
BGD	0.023966	IRQ	0.08467	RWA	0.0625
BLZ	0.073643	ISR	0.343333	SAH	1
BOL	0.068859	JAM	0.089431	SAU	0.039423
BRA	0.092262	JOR	0.032558	SDN	0.213807
BRN	0.16129	KAB	0.055046	SDS	0.347961
BTN	0.039623	KAS	1	SEN	0.033037
BWA	0.002074	KAZ	0.064847	SGP	1
CAF	0.064724	KEN	0.018052	SLE	0.285542
CHL	0.120278	KGZ	0.159517	SLV	0.057613
CHN	0.119058	KHM	0.091589	SLO	0.15408
CIV	0.049419	KWT	0.081448	SOM	0.05876
CMR	0.047654	LAO	0.059844	SUR	0.110259
CNM	0.333333	LBN	0.086331	SWZ	0.40708
COD	0.079909	LBR	0.016187	SYR	0.034994

COG	0.14932	LBY	0.019029	TCD	0.019425
COL	0.16678	LKA	0.087649	TGO	0.161919
CRI	0.1198	LSO	0.779412	THA	0.089627
CUB	0.130506	MAR	0.069437	TJK	0.112322
CYN	0.566667	MDG	0.095638	TKM	0.026463
CYP	0.352941	MEX	0.067058	TLS	0.108974
DJI	0.4375	MLI	0.066136	TTO	0.212766
DOM	0.062609	MMR	0.102656	TUN	0.031632
DZA	0.027011	MNG	0.094879	TUR	0.084819
ECU	0.1261	MOZ	0.055876	TWN	0.129841
EGY	0.030746	MRT	0.032622	TZA	0.093735
ERI	0.011065	MWI	0.097397	UGA	0.090836
ESB	0	MYS	0.107326	URY	0.093725
ETH	0.034781	NAM	0.016376	UZB	0.098051
GAB	0.026928	NER	0.025251	VEN	0.086985
GEO	0.136406	NGA	0.100411	VNM	0.120401
GHA	0.050751	NIC	0.063336	YEM	0.020265
GIN	0.140271	NPL	0.059487	ZAF	0.039901
GMB	0.210084	OMN	0.119071	ZMB	0.088797
				ZWE	0.017105

Minor points:

- Line 35: given that non-therapeutical use of antimicrobials is (slowly, but still) decreasing, please add something like "in certain regions of the world" or similar at the end of the sentence

Thank you for the suggestion. We added "in some regions of the world" at the end of the sentence:

"In animals, antimicrobials are used for treatment but also as surrogates for good hygiene practices and to increase productivity on farm in some regions of the world²."

- Lines 43-45: the reference here (9) is based on data from Canada, so perhaps you could have "North America" or "US and Canada" instead of just US at the beginning of the sentence in line 43. Furthermore, ceftiofur is a drug and not an antimicrobial class, so consider having "e.g. cephalosporins such as ceftiofur" or similar given that the sentence seems to be based on "limitation of certain classes of antimicrobials".

Thank you for the suggestion. We added Canada along with the US at the beginning of the sentence.

"In high-income settings such as the US⁷, Canada⁸, and the EU countries⁹, animal AMR has been the focus of systematic surveillance for decades."

- Lines 66-69: the authors speculate on the usefulness of the identification of antimicrobials for which resistance will go above a certain level (e.g., >50%) in a given region in the future as a tool to "inform veterinarians on possible treatment options". While I agree in principle, in order for this to happen in reality it would be necessary to measure resistance considering clinical breakpoints adapted to animals, since while a treatment is still effective in the field veterinarians and farmers are likely to still use it (even if the resistance according to CLSI breakpoints is above 50 or 80%). It would be good to add somewhere in the discussion the need for considering animal breakpoints if animal health is going to be a driver of veterinary AMU. Furthermore, the two bacteria selected here (*E. coli* and *Salmonella*) can be often found in apparently healthy animals and presumably a large proportion of the references used here were based on sampling of non-diseased animals. Therefore, some discussion on the need to consider other (not necessarily zoonotic but still relevant!) animal pathogens in this kind of exercises (if the point is to help "inform veterinarians on possible treatment options" would be useful to advance towards a real one health approach to the issue of AMR (and AMU). I agree that having a high resistance to a given antimicrobial in e.g. commensal *E. coli* is certainly signaling a problem, but this may have only limited implications in terms of treatment if 1) diseases are mediated by other pathogens and 2) treatment can be still successful as currently performed.

We agree with the reviewer that the identification of antimicrobials for which resistance will go above a certain level can only have limited use for informing veterinarians on possible treatment options. All surveys that are included in the current analyses were conducted on healthy animals, and the majority of surveys used human clinical breakpoints for determining resistance phenotypes. We revised the sentences on line 67-70 from

“Having the ability to predict which antimicrobials will cross critical resistance levels in the future could inform veterinarians on possible treatment options, as well as strengthening local surveillance efforts.”

to the following:

“Having the ability to predict which antimicrobials will cross critical resistance levels in the future could help assess the risk of antimicrobial resistant infections acquired from animal sources, as well as strengthening local surveillance effort.”

We also added the following in the discussion section:

“Our prediction of priority antimicrobials was based on surveys conducted exclusively on commensal *E. coli* and *Salmonella* from healthy animals, and the majority of surveys used human clinical breakpoints to determine resistance phenotype. However, our approach could also be adapted to databases of AMR of other animal pathogens using veterinary clinical breakpoints, to help inform veterinarians on possible treatment options in regions of high AMR levels.”

Reviewer #3 (Remarks to the Author):

Overall, this is a welcome addition to the work on AMR in food animals in LMICs as the previously reported P50 measure is not as actionable as presenting estimates for individual antibiotics or antibiotic classes. There's a lot of value in describing the current prevalence of resistance and identifying hotspots where intensified efforts at slowing the progression of resistance are needed. The estimates rely on robust data from point prevalence surveys with more limited geographic coverage in parts of South America and central-sub Saharan Africa. There are appropriate adjustments for changes in breakpoints across time, methodology, and standard used.

There are a number of questions and concerns that are mostly driven by the estimates related to forecasting: The forecasting methods are unclear. The independent variables for the models end at 2019 or 2020 so what is being used to inform the prediction? What, precisely, is the dependent variable and what's informing the time to resistance exceeding 50% estimate? Does the point estimate of prevalence of resistance for a given pixel inform this model at all or is it only the dichotomized resistance above or below 50%?

We thank the reviewer for pointing out that the explanations on the forecasting methods were indeed too succinct. The reviewer is correct that we cannot have covariates value for the future. The model thus uses spatial variations of resistance profiles for predicting temporal changes of resistance profiles. Using the example shown in Figure 5, assume one location on the map has current resistance profile rp_0 , and we are trying to predict the probability that CIP will be the next drug with $>50\%$ resistance (resistance profile rp_1) in the near future. To do this, we extract covariates associated with locations of PPS that have reported resistance profiles of rp_1 (presence), rp_2 and rp_3 (absences), and these are used as independent variables to train the model. Additionally, due to the existence of co-resistance between antimicrobials, the probability of the occurrence of rp_1 , rp_2 , or rp_3 are not identical, and this is captured by using the proportion of surveys reporting rp_1 out of all three resistance profiles as independent variables in the model. The dependent variable is binarized as 1 (presence) for rp_1 , and 0 (absence) for rp_2 and rp_3 . Only the dichotomized resistance above or below 50% for a given pixel informs the model, not the exact value of the prevalence of resistance. However, we also conducted the analysis using 10% and 25% as the cutoff values to assess the sensitivity of our findings.

We have provided a revised explanation of our model in lines 496 – 508 based on our response to referee #3:

“The covariates used to predict its presence and absence included two components. The first component considers patterns of co-resistance between antimicrobials, implying that probabilities of occurrence vary between resistance profiles. This variation is captured by using the proportion of surveys recording rp_1 out of all surveys recording rp_1 , rp_2 or rp_3 as a covariate (Figure 5d.i). Patterns of co-resistance also implies that the development of resistance of CIP is dependent on resistance of other antimicrobials. This dependence is captured by using the number of antimicrobials with resistance higher than 50% in the resistance profile in 2019 as a covariate (Figure 5d.ii). The second component of covariates includes risk factors for predicting the development of resistance. This includes the percentage of CIP use (kg) out of all three antimicrobials at the location of the survey (Figure 5d.iii), as well as a set of environmental and anthropogenic covariates associated with the locations of the surveys, such as total antimicrobial use in 2013 and 2020, temperature, and animal density (Figure 5d.iv; Supplementary Table 3).”

The result of the forecasting is surprising, the map in figure 3A suggests that tetracyclines are the antimicrobial with the highest probability of >50% resistance for most pixels but the time until resistance exceeds 50% is 5.1 years for tetracyclines compared to 1.7 years for penicillins and 4.1 years for cephalosporins.

The reason that the time it takes for resistance to exceed 50% for penicillins and cephalosporins is not necessarily higher than tetracycline is because they are associated with different pixels. For example, in pixels where cephalosporins was the priority antimicrobial, resistance to tetracycline has mostly already exceeded 50%.

In table S2 why are aminoglycosides NA?

Supplementary Table 4 (previously numbered Supplementary Table 2) had NA value for aminoglycosides because there was no pixel where aminoglycosides was predicted as the priority antimicrobial (antimicrobial with the highest probability of its resistance prevalence exceeding 50% in the near future).

The trends of AMR by antimicrobial class using linear regression for tetracycline and ampicillin is a little concerning and is conceptually a bit tough to understand why these two antibiotics in particular would violate linearity using the logit of the dependent variable; for example, the ampicillin is being modeled with a linear regression when the cefotaxime is logistic regression, this is surprising since they're broadly part of the same antibiotic class - i.e. both beta lactam antibiotics and mechanism of resistance often leads to overlapping resistance and probably deserves some further comment or exploration as to why this is plausible. It's also noteworthy that tetracycline and ampicillin are the two antibiotics most commonly identified by the authors as priority antibiotics and also happen to utilize a model distinct from the other 5 antimicrobials.

Thank you for raising this remark, which helps us identify a statistical anomaly. We agree with the reviewer that using two different types of model could bring issues of comparability between predictions. Therefore, we investigated the reason for the violation of the linearity assumption for AMP and TET, and identified that it was caused by one single outlier survey in 2000 that reported very high value for the resistance prevalence of AMP and TET in India (100% resistance, DOI: 10.1264/jsme2.2000.173). This represents one PPS out of 758 PPS that reported resistance for TET, and 797 PPS that reported resistance for AMP. We have removed this outlier, and used logistic regression for all antimicrobials.

We added explanation on the removal of outlier in the Method section in lines 429 – 433:

“For TET and AMP, we removed one outlier (DOI of PPS: 10.1264/jsme2.2000.173) out of 758 PPS reporting resistance for TET and 797 PPS reporting resistance for AMP, to ensure that the assumption of linearity between the logit of dependent variable and the independent variable was met based on results of Box-Tidwell test.”

I would be interested to know how the candidate covariates were selected – e.g. percentage of tree cover.

The candidate covariates were selected as they may have plausible relationship with either antimicrobial use, or the transmission of antimicrobial resistance. Percentage of tree coverage, pesticide application rate, atmospheric ammonia are proxies for the degree of agriculture intensification and therefore associated with antimicrobial use. Travel time to cities may be linked with accessibility to antimicrobials from drug stores. Temperature may be linked with animal injuries and therefore antimicrobial application. Irrigation may be linked with the spread of antimicrobial resistance genes. Pesticide application rate, atmospheric ammonia, and percentage of tree coverage are proxies for the degree of agriculture intensification and therefore associated with antimicrobial use. GDP has been shown to be associated with changes with antimicrobial consumption rate.

Can the authors comment on the coefficients for the proportion_PPS covariate being positive for some antimicrobials and negative for others?

We think that a plausible reason for some coefficients for “proportion_PPS” to be negative is due to the association (and possibly collinearity) between “proportion_PPS” and “N50” – the number of antimicrobials with >50% resistance in the resistance profile (covariate ii in Figure 5). When investigated individually, “proportion_PPS” has positive coefficients for all antimicrobials. When both covariates are included, coefficients of “proportion_PPS” decreased to negative in some cases with high values of “N50”.

I assume the proportion_PPS covariate is what's described in figure5 (d)[i] if so I would seek clarification on whether a profile where CIP, GEN, and CTX are all >50% resistance would be included in rp1 (ie in the numerator) or excluded altogether. I assume it's the former but it's not clear.

The reviewer's interpretation is correct that a resistance profile with >50% resistance for all of CIP, GEN, and CTX was not included in rp1. Our model is only looking at scenarios where one additional drug – in comparison with rp0 – will exceed 50% resistance in the future. We added an explanation on this in the Method section in lines 491 - 493:

“The model considers future scenarios where only one additional antimicrobial will exceed 50% resistance (Figure 5b).”

Is proportion_PPS location specific? If a location in India and Brazil have the same current resistance profile (rp) would they also have the same proportion_PPS? The labeling in figure 5 might need to be reviewed, particularly the second column ‘future resistance profile,’ it’s not clear that a temporal relationship is being enforced in the production of the proportion_PPS covariate.

Proportion_PPS is not location specific. The reviewer is right that if two locations in India and Brazil have the same current resistance profile (rp0), they would also have the same proportion_PPS for the future resistance profiles. Values of proportion_PPS was limited in America and Africa, and region-specific proportion_PPS may be possible in the future as the amount of data steadily increases in these two regions.

We revised “current resistance profile” to “resistance profile in 2015” (2015 was the median year of publication for PPS that were used for producing our maps of resistance prevalence), and revised “future resistance profile” to “subsequent resistance profiles in the near future”.

Figure 2 should include a year, I would also advocate for including all 7 antimicrobials in the figure.

Thank you for the suggestion, we added the years in the caption, and included all 7 antimicrobials.

Figure 2. Geographic distribution of antimicrobial resistance between 2000 and 2019. Prevalence of resistance (Prev. Res.) in *E. coli* and *Salmonella* for tetracycline (A,C), ampicillin (B, D), sulfamethoxazole-trimethoprim (E, G), chloramphenicol (F, H), ciprofloxacin (I, K), gentamicin (J, L), cefotaxime (M, O). Number of antimicrobials (out of 7) with resistance higher than 50% (N50; N, P) (See Supplementary Figure 7 for maps generated using cutoff values other than 50%). Maps of resistance prevalence for the 7 antimicrobials are available on resistancebank.org.

The work this seems to build on produced estimates by animal type, this work moved away from that and didn't seem to include every animal type as covariates according to table S3. I'd welcome the author's comments on that shift since it seemed like they identified resistance rising in chickens faster than other food animals.

Thank you for suggesting to include estimates by animal type. This was initially not included because the manuscript was intended to be focused on the breakdown by antimicrobial classes. We now included estimates of resistance trends per animal type in Supplementary Figures 1-3 (see below).

The reason that Supplementary Table 5 (previously numbered Supplementary Table 3) does not include all animal types was because we excluded rows where all coefficients was 0 based on LASSO regression, and these included the population density of pigs and cattle. We added the explanation in the legend of Supplementary Table 5: “Covariates for which coefficients were 0 for all antimicrobials were removed from the table.”

Supplementary Figure 1. **Chicken:** temporal trends of the prevalence of resistance, for ampicillin (AMP), chloramphenicol (CHL), ciprofloxacin (CIP), cefotaxime (CTX), gentamicin (GEN), sulfamethoxazole-trimethoprim (SXT), and tetracycline (TET). Solid lines represent significant temporal trends ($p < 0.05$), and dashed lines represent nonsignificant trends. Transparency levels of the red colors were proportional to the number of surveys published each year. Temporal trends were significant (p value < 0.05) for AMP, CHL, CIP, and CTX.

Supplementary Figure 2. **Pigs:** Temporal trends of the prevalence of resistance, for ampicillin (AMP), chloramphenicol (CHL), ciprofloxacin (CIP), cefotaxime (CTX), gentamicin (GEN), sulfamethoxazole-trimethoprim (SXT), and tetracycline (TET). Solid lines represent significant temporal trends ($p < 0.05$), and dashed lines represent nonsignificant trends. Transparency levels of the red colors were proportional to the number of surveys published each year. Temporal trends were significant (p value < 0.05) for all antimicrobials except TET.

Supplementary Figure 3. **Cattle:** temporal trends of the prevalence of resistance in cattle, for ampicillin (AMP), chloramphenicol (CHL), ciprofloxacin (CIP), cefotaxime (CTX), gentamicin (GEN), sulfamethoxazole-trimethoprim (SXT), and tetracycline (TET). Solid lines represent significant temporal trends ($p < 0.05$), and dashed lines represent nonsignificant trends. Transparency levels of the red colors were proportional to the number of surveys published each year. Temporal trends were not significant ($p > 0.05$) for all antimicrobials.

I appreciate the representation of uncertainty but it would also be helpful to know the number of isolates informing the estimates so that readers can better appreciate the limitations or strengths of the data. Supp 9b number of isolates, in the legend add average number of isolates per survey per region.

Thank you for suggesting to map the number of isolates informing the estimates. We added Supplementary Figure 14 which showed the number of isolates used to determine resistance prevalence in each PPS.

Supplementary Figure 14. Geographic locations of point-prevalence surveys reporting resistance prevalence of *E. coli* and *Salmonella*. Sizes of the circle were in proportion to the log₁₀ transformed sample sizes of each survey. Colors of the circles represented the number of bacterial isolates used to test the prevalence of resistance in each survey. The average number of isolates in each survey was 71 in Africa, 98 in America, and 94 in Asia.

The discussion focuses a lot on priorities for surveillance and not enough on measures to reduce antimicrobial use in food animals.

Thank you for pointing out the lack of discussion on measures to reduce antimicrobial use. We now added a paragraph in lines 231 – 239:

“Measures to contain AMR in the identified hotspot regions will need to be focused on reducing antimicrobial use as well as strengthening biosecurity in farms. Enforcing a regulation with a cap of 50 milligram antimicrobial used per kilogram of food animal products was estimated to reduce global antimicrobial consumption by 64%¹. However, major investment on the surveillance of antimicrobial use is needed for such regulations to be effective. Improving biosecurity in farms may reduce the reliance on antimicrobials for keeping the animals healthy. Measures to improve biosecurity include stricter hygienic control on farm entry and better separation between compartments in the farm, and can be facilitated by risk-based quantitative tools¹⁹.”

REVIEWERS' COMMENTS

Reviewer #1 (Remarks to the Author):

- What are the noteworthy results?

Building on previous work, this work provides some insight into the burden of AMR in animals for two important bacteria and seven antibacterials. This work could inform local surveillance systems to help them understand the burden of AMR in their animal populations.

- Will the work be of significance to the field and related fields? How does it compare to the established literature? If the work is not original, please provide relevant references.

This work builds on previous work the group have completed.

- Does the work support the conclusions and claims, or is additional evidence needed?

Additional data would be helpful, but this requires more, ideally, prospective studies

- Are there any flaws in the data analysis, interpretation and conclusions? Do these prohibit publication or require revision?

I don't see any major flaws in the data analysis, interpretation or conclusions.

- Is the methodology sound? Does the work meet the expected standards in your field?

The methodology is sound and has been used by the authors previously.

- Is there enough detail provided in the methods for the work to be reproduced?

Yes, it also looks as though the data and the code shared.

In more detail:

The manuscript is much improved, however I do have some specific comments.

Overall comments:

- Are the grey areas on your maps areas that have not been modelled (for example many parts of Europe and Australia).
- It would be useful to say why you chose the seven antibacterials and which infections are the bacteria used for in animals to understand their importance and the problems we will have if we are no longer able to use any of the antibiotics listed (eg if we no longer had tetracycline we would not be able to treat mycoplasma etc)

Abstract:

- What do you mean when you say your maps will "provide a baseline for targeting resources for AMR surveillance"? Please explain this in full

Introduction:

- You mention the World Health Organization's list of Critically Important Antimicrobials (CIA), this has now been updated to the list of Medically Important Antimicrobials (MIA; https://cdn.who.int/media/docs/default-source/antimicrobial-resistance/amr-gcp-irc/who_mialist_draft_forexternaldiscussion.pdf?sfvrsn=af6f2ebf_1), no reference to either documents can be found in the list of references.
- Write *Escherichia coli* and *Salmonella* species in the first instance
- Are you describing *Salmonella* species or nontyphoidal *Salmonella* species? There is mention of nontyphoidal *Salmonella* at one point, if this is this the species throughout please use the full name. The two species are very different and should not be confused.

Results

- Add the numbers and denominators to the percentage to ensure transparency in your results section
- Figure 2 is difficult to interpret, can this be made more clear?

Discussion

- You should qualify that the data in your previous paper (Pires et al. 2019) was included in the current publication (therefore it's not a surprise that the results are consistent with your previous finding)
- Within your priority antimicrobials for AMR surveillance paragraph you describe your computational approach to map priority antimicrobials for surveillance incorporating dependencies on local risk factors. This seems very important to me and could be an impactful way of reducing

the use of antibiotics or changing the ones that are in use locally, I suggest you bring this more to the forefront in the paper

Supplementary information:

- It would be useful to see the results of the comparison between the three imputation methods (LASSO-GLM, BLR and NN)
- Summary table 11 – I assume 0.21 means 21% of the 10x20km pixels in Afghanistan have uncertainty above 40% (and in this table that 1 is 100%) – please clarify and add an explanation to the table

References:

Double check all of the references as there was no reference for the WHO MIA (which should have been reference 14).

Reviewer #2 (Remarks to the Author):

The responses provided by the authors and changes introduced in the manuscript have improved significantly the clarity of the text, and strengths and limitations of data/analysis are now more evident and easy to interpret.

Reviewer #3 (Remarks to the Author):

I thank the authors for the revisions to the text, they provide a bit more clarity on the methods. I have a few further points that remain unclear to me.

The map in figure 2 title note it's showing resistance between 2000 and 2019, but it seems like one of the core arguments of the paper is that resistance is increasing across time. So what is being shown in figure 2? Is it a mean or median of resistance for each pixel across two decades?

Figure S10(A) and Sup table 4 present time in years for resistance to exceed 50% but it's unclear how this value is being estimated. If I understand it correctly figure 5 describes subsequent resistance profile 'in the near future' to determine which antibiotic is most likely to exceed 50% next, I don't see where time in years for that to occur is predicted.

I have some concerns about the dichotomization of prevalence of resistance to <50% and >50% (or 10%, 25%) as it would seem proximity to that threshold would be at least as important as patterns of resistance profiles in other locations in terms of informing the probability that the antibiotic in question would pass that threshold. These concerns are somewhat assuaged by the AUC cross validation but feel it at least deserves mention in limitations.

Response to reviewers #NCOMMS-23-2661A

The comments from the reviewers are in **black**, the responses to reviewers and modifications to the original manuscript are in **blue**, quotes to the original manuscript are provided in **grey**.

Reviewer #1 (Remarks to the Author):

- What are the noteworthy results?

Building on previous work, this work provides some insight into the burden of AMR in animals for two important bacteria and seven antibacterials. This work could inform local surveillance systems to help them understand the burden of AMR in their animal populations.

- Will the work be of significance to the field and related fields? How does it compare to the established literature? If the work is not original, please provide relevant references.

This work builds on previous work the group have completed.

- Does the work support the conclusions and claims, or is additional evidence needed?

Additional data would be helpful, but this requires more, ideally, prospective studies

- Are there any flaws in the data analysis, interpretation and conclusions? Do these prohibit publication or require revision?

I don't see any major flaws in the data analysis, interpretation or conclusions.

- Is the methodology sound? Does the work meet the expected standards in your field?

The methodology is sound and has been used by the authors previously.

- Is there enough detail provided in the methods for the work to be reproduced?

Yes, it also looks as though the data and the code shared.

In more detail:

The manuscript is much improved, however I do have some specific comments.

Overall comments:

- Are the grey areas on your maps areas that have not been modelled (for example many parts of Europe and Australia).

The reviewer is correct that the grey areas on the maps are areas that were not included in our model outputs: our analysis focused on low- and middle-income countries (LMICs) where systematic surveillance is largely absent. We intentionally did not include high-income countries in the analysis. We added an explanation for our focus on LMICs in all figure legends:

“Figure 1. Temporal trends of the prevalence of resistance in low- and middle-income countries for ampicillin (AMP), chloramphenicol (CHL), ciprofloxacin (CIP), cefotaxime (CTX), gentamicin (GEN), sulfamethoxazole-trimethoprim (SXT), and tetracycline (TET).”

“Figure 2. Geographic distribution of antimicrobial resistance in low- and middle-income countries between 2000 and 2019.”

“Figure 3. A) Geographic distribution of antimicrobials with the highest probability of their resistance prevalence exceeding 50% in the future in low- and middle-income countries.”

- It would be useful to say why you chose the seven antibacterials and which infections are the bacteria used for in animals to understand their importance and the problems we will have if we are no longer able to use any of the antibiotics listed (eg if we no longer had tetracycline we would not be able to treat mycoplasma etc)

Thank you for the suggestion. We followed the referee's feedback and added a justification for focusing on 7 classes of antimicrobials and their importance for treating animal diseases in lines 207 – 213:

“The 7 antimicrobial classes included in the analysis are the most frequently cited classes across 1,088 point prevalence survey, and are important for treating infectious diseases in food animals. For example, tetracycline is widely used for treating *Mycoplasma* in chicken¹⁹, gentamicin is used for treating *Pseudomonas aeruginosa* infections²⁰, and third- and fourth-generation cephalosporins are used for treating cattle mastitis²⁰. Therefore, rising resistance levels in these drugs may lead to therapy failure, and thereby negatively impact animal health and the agricultural economy.”

Abstract:

- What do you mean when you say your maps will “provide a baseline for targeting resources for AMR surveillance”? Please explain this in full

We have now revised the sentence from

“Our maps provide a baseline for targeting resources for AMR surveillance, and adapting policies to local epidemiological context across LMICs.”

to

“Our maps highlight diverging geographic trends of AMR prevalence across antimicrobial classes, and can be used to target AMR surveillance in AMR hotspots for priority antimicrobial classes.

Introduction:

- You mention the World Health Organization’s list of Critically Important Antimicrobials (CIA), this has now been updated to the list of Medically Important Antimicrobials (MIA; https://cdn.who.int/media/docs/default-source/antimicrobial-resistance/amr-gcp-irc/who_mialist_draft_forexternaldiscussion.pdf?sfvrsn=af6f2ebf_1), no reference to either documents can be found in the list of references.

Thank you for pointing out the update on the reference. We corrected it to the list of WHO “MIA”, and revised “Critically Important Antimicrobials” to “Medically Important Antimicrobials” throughout the manuscript.

- Write *Escherichia coli* and *Salmonella* species in the first instance

Thank you, we now mentioned “*Escherichia coli* and nontyphoidal *Salmonella* species” in the abstract as well as in the first instance in introduction.

- Are you describing *Salmonella* species or nontyphoidal *Salmonella* species? There is mention of nontyphoidal *Salmonella* at one point, if this is the species throughout please use the full name. The two species are very different and should not be confused.

Thank you. We are referring to nontyphoidal *Salmonella*, and have added the full name throughout the manuscript.

Results

- Add the numbers and denominators to the percentage to ensure transparency in your results section

Thank you for the suggestion. We added the number of samples/surveys behind the percentages in the results section:

“The mean prevalence of resistance weighted by the number of samples in each PPS, in *E. coli* and nontyphoidal *Salmonella*, was respectively 59% (n = 745) and 54% (n = 597) for tetracycline (TET), 57% (n = 779) and 46% (n = 632) for ampicillin (AMP), 45% (n = 649) and 36% (n = 501) for sulfamethoxazole-trimethoprim (SXT), 35% (n = 656) and 26% (n = 553) for chloramphenicol (CHL), 30% (n = 796) and 26% (n = 624) for ciprofloxacin (CIP), 28% (n = 882) and 23% (n = 650) for gentamicin (GEN), and 33% (n = 446) and 19% (n = 334) for cefotaxime (CTX). Between 2000 and 2019, changes in the prevalence of resistance were +12% (TET), +33% (AMP), +19% (SXT), +20% (CHL), +16% (CIP), +11% (GEN), and +37% (CTX) (Figure 1). The temporal increases of resistance were significant ($p < 0.05$) for all antimicrobials apart from TET.

Prevalence of resistance was investigated in poultry in 52% (n = 570) of PPS, in cattle in 38% (n = 409) of PPS, and in pigs in 28% (n = 303) of PPS. Prevalence of resistance increased significantly for AMP, CHL, CIP, and CTX for poultry, and for AMP, SXT, CHL, CIP, GEN, and CTX for pigs (Supplementary Figures 1 to 2). However, temporal trends of resistance were not significant for any antimicrobial classes for cattle (Supplementary Figures 3).”

- Figure 2 is difficult to interpret, can this be made more clear?

Thank you for pointing out the lack of clarity in Figure 2, which we think is mainly due to too many panels presented in the same figure. We now separated this figure into one figure for each bacterium: figure 2 for *E. coli* and figure 3 for *Salmonella*.

Figure 2. Geographic distribution of antimicrobial resistance in *E. coli* in low- and middle-income countries between 2000 and 2019 (median year 2015). Prevalence of resistance (Prev. Res.) for tetracycline (a), ampicillin (b), sulfamethoxazole-trimethoprim (c), chloramphenicol (d), ciprofloxacin (e), gentamicin (f), cefotaxime (g). Overall resistance level across antimicrobials measured using the number of antimicrobials (out of 7) with resistance higher than 50% (N50; h) (See Supplementary Figure 7 for maps generated using cutoff values other than 50%). Maps of resistance prevalence for the 7 antimicrobials are available on resistancebank.org.

Figure 3. Geographic distribution of antimicrobial resistance in nontyphoidal *Salmonella* in low- and middle-income countries between 2000 and 2019 (median year 2015). Prevalence of resistance (Prev. Res.) for tetracycline (a), ampicillin (b), sulfamethoxazole-trimethoprim (c), chloramphenicol (d), ciprofloxacin (e), gentamicin (f), cefotaxime (g). Overall resistance level across antimicrobials measured using the number of antimicrobials (out of 7) with resistance higher than 50% (N50; h) (See Supplementary Figure 7 for maps generated using cutoff values other than 50%). Maps of resistance prevalence for the 7 antimicrobials are available on resistancebank.org.

Discussion

- You should qualify that the data in your previous paper (Pires et al. 2019) was included in the current publication (therefore it's not a surprise that the results are consistent with your previous finding)

We added the following in lines 182 – 184:

“This consistency can be partly attributed to the incorporation of a subset of PPS used in Van Boeckel & Pires et al. 2019 into the present analysis (Supplementary Information).”

- Within your priority antimicrobials for AMR surveillance paragraph you describe your computational approach to map priority antimicrobials for surveillance incorporating dependencies on local risk factors. This seems very important to me and could be an impactful way of reducing the use of antibiotics or changing the ones that are in use locally, I suggest you bring this more to the forefront in the paper

Thank you for the suggestion. We moved section “*Priority antimicrobials for AMR surveillance*” earlier in the discussion, below section “*Geographic distribution of AMR*”.

Supplementary information:

- It would be useful to see the results of the comparison between the three imputation methods (LASSO-GLM, BLR and NN)

Thank you for the suggestion. We produced maps of priority antimicrobials for AMR surveillance using the three imputation methods in Supplementary Figure 17. Compared with LASSO-GLM (Supplementary Figure 17a), the

predicted priority antimicrobials were different in 0.06% of the pixels in the map produced using BLR (Supplementary Figure 17b), and in 2.6% of pixels in the map produced using NN (Supplementary Figure 17c).

Supplementary Figure 17. Geographic distribution of antimicrobials with the highest probability of their resistance prevalence exceeding 50% in the future, with missing resistance prevalence in each survey imputed using LASSO regression (a), Bayesian linear regression (b), and feed-forward neural network (c). TET: tetracycline; AMP: ampicillin; SXT: sulfamethoxazole-trimethoprim; CHL: chloramphenicol; CIP: ciprofloxacin; GEN: gentamicin; CTX: cefotaxime.

• Summary table 11 – I assume 0.21 means 21% of the 10x20km pixels in Afghanistan have uncertainty above 40% (and in this table that 1 is 100%) – please clarify and add an explanation to the table

Thank you for pointing out the lack of clarity in the presented numbers. We have now revised the format of the numbers into percentages, and revised the legend of Supplementary Table 11. We hope this provides more clarity now:

Supplementary Table 11. The percentage of 10 x 10 km pixels in each country that have an uncertainty of the predicted priority antimicrobial above 40%.

Country ISO3	Percentage of pixels with uncertainty > 40%	Country ISO3	Percentage of pixels with uncertainty > 40%	Country ISO3	Percentage of pixels with uncertainty > 40%
AFG	21%	GNB	16%	PAK	8%
AGO	3%	GNQ	0%	PAN	11%
ARE	15%	GTM	10%	PER	14%
ARG	3%	GUY	12%	PHL	16%
ARM	15%	HND	3%	PRI	6%

AZE	6%	HTI	9%	PRK	77%
BDI	7%	IDN	14%	PRY	3%
BEN	8%	IND	7%	PSX	38%
BFA	6%	IRN	13%	QAT	16%
BGD	2%	IRQ	8%	RWA	6%
BLZ	7%	ISR	34%	SAH	100%
BOL	7%	JAM	9%	SAU	4%
BRA	9%	JOR	3%	SDN	21%
BRN	16%	KAB	6%	SDS	35%
BTN	4%	KAS	100%	SEN	3%
BWA	0%	KAZ	6%	SGP	100%
CAF	6%	KEN	2%	SLE	29%
CHL	12%	KGZ	16%	SLV	6%
CHN	12%	KHM	9%	SLO	15%
CIV	5%	KWT	8%	SOM	6%
CMR	5%	LAO	6%	SUR	11%
CNM	33%	LBN	9%	SWZ	41%
COD	8%	LBR	2%	SYR	3%
COG	15%	LBY	2%	TCD	2%
COL	17%	LKA	9%	TGO	16%
CRI	12%	LSO	78%	THA	9%
CUB	13%	MAR	7%	TJK	11%
CYN	57%	MDG	10%	TKM	3%
CYP	35%	MEX	7%	TLS	11%
DJI	44%	MLI	7%	TTO	21%
DOM	6%	MMR	10%	TUN	3%
DZA	3%	MNG	9%	TUR	8%
ECU	13%	MOZ	6%	TWN	13%
EGY	3%	MRT	3%	TZA	9%
ERI	1%	MWI	10%	UGA	9%
ESB	0%	MYS	11%	URY	9%
ETH	3%	NAM	2%	UZB	10%
GAB	3%	NER	3%	VEN	9%
GEO	14%	NGA	10%	VNM	12%
GHA	5%	NIC	6%	YEM	2%
GIN	14%	NPL	6%	ZAF	4%
GMB	21%	OMN	12%	ZMB	9%
				ZWE	2%

References:

Double check all of the references as there was no reference for the WHO MIA (which should have been reference 14).

Thank you, we have now corrected this reference.

Reviewer #2 (Remarks to the Author):

The responses provided by the authors and changes introduced in the manuscript have improved significantly the clarity of the text, and strengths and limitations of data/analysis are now more evident and easy to interpret.

Thank you.

Reviewer #3 (Remarks to the Author):

I thank the authors for the revisions to the text, they provide a bit more clarity on the methods. I have a few further points that remain unclear to me.

The map in figure 2 title note it's showing resistance between 2000 and 2019, but it seems like one of the core arguments of the paper is that resistance is increasing across time. So what is being shown in figure 2? Is it a mean or median of resistance for each pixel across two decades?

In Figure 2, ideally, to obtain a map depicting temporal trends, we would have liked to be able to use a spatio-temporal model instead of only spatial model for one mid-point in time. However, this is currently challenged by the limited amount of data available to reach the necessary statistical power to fit a spatio-temporal model. In such case, we think the median year (2015) of the PPS is good approximation for the year associated with the predicted values in each pixel. Here, median is preferred than mean, because the years of the PPS are skewed towards more recent years, and median can better capture the central tendency in such case. We added a point to the discussion section to reflect the limitations associated with having to choose a mid-point in time to summarize the temporal trends on the point prevalence surveys (lines 312 – 323):

“Secondly, due to the limited number of surveys reporting resistance prevalence for individual antimicrobial-bacteria combinations, mapped predictions of AMR were restricted to 7 drugs and 2 bacteria. These drugs were amongst the most frequently used antimicrobial classes and the most frequently cited classes across 1,088 point prevalence surveys. Additionally, predictions of nontyphoidal *Salmonella* were not disaggregated for individual serovars. However, this is in consistency with Murrey et al. 2022 who mapped AMR in humans¹². **The limited number of surveys available also made it challenging to conduct spatio-temporal modelling, and we pooled together surveys from all years for AMR mapping.** As the number of point prevalence surveys³⁴ published each year is growing, future efforts to map AMR may incorporate more antimicrobial-bacteria combinations **and investigate both spatial and temporal effects on AMR maps**, while insuring statistical robustness in the extrapolations.”

We also added the median year of 2015 in the legends for Figure 2 and Figure 3 (note that we now separate Figure 2 into two figures for better clarity):

Figure 2. **Geographic distribution of antimicrobial resistance in *E. coli* in low- and middle-income countries between 2000 and 2019 (median year 2015).** Prevalence of resistance (Prev. Res.) for tetracycline (a), ampicillin (b), sulfamethoxazole-trimethoprim (c), chloramphenicol (d), ciprofloxacin (e), gentamicin (f), cefotaxime (g). Overall resistance level across antimicrobials measured using the number of antimicrobials (out of 7) with resistance higher than 50% (N50; h) (See Supplementary Figure 7 for maps generated using cutoff values other than 50%). Maps of resistance prevalence for the 7 antimicrobials are available on resistancebank.org.

Figure 3. **Geographic distribution of antimicrobial resistance in nontyphoidal *Salmonella* in low- and middle-income countries between 2000 and 2019 (median year 2015).** Prevalence of resistance (Prev. Res.) for tetracycline (a), ampicillin (b), sulfamethoxazole-trimethoprim (c), chloramphenicol (d), ciprofloxacin (e), gentamicin (f), cefotaxime (g). Overall resistance level across antimicrobials measured using the number of antimicrobials (out of 7) with resistance higher than 50% (N50; h) (See Supplementary Figure 7 for maps generated using cutoff values other than 50%). Maps of resistance prevalence for the 7 antimicrobials are available on resistancebank.org.

Figure S10(A) and Sup table 4 present time in years for resistance to exceed 50% but it's unclear how this value is being estimated. If I understand it correctly figure 5 describes subsequent resistance profile 'in the near future' to determine which antibiotic is most likely to exceed 50% next, I don't see where time in years for that to occur is predicted.

This estimation was calculated separately from the process described in Figure 5. Figure 5 described the procedure to predict which antimicrobial will most likely exceed 50% resistance in the future. Based on these predictions, we then used the temporal logistic regression model of the corresponding antimicrobial, as shown in Figure 1, to estimate the time it takes for resistance to rise from the current predicted level to 50%.

This process was originally described at the end of the Method section "Mapping priority antimicrobials for AMR surveillance":

“Additionally, at each pixel on the map of priority antimicrobials for AMR surveillance, we estimated the time necessary for the resistance prevalence of the corresponding priority antimicrobial to reach 50% in the future. Concretely, we extracted the current resistance prevalence estimated at each pixel, and calculated the time difference from the current resistance prevalence until it reaches 50%, using the corresponding regression models fitted in section “Trends of AMR for each antimicrobial class”.”

We now revised this description in lines 504 – 510, and provided an illustration in Supplementary Figure 16 to demonstrate the calculation:

“Furthermore, based on predictions of the priority antimicrobial for AMR surveillance at each 10x10 km pixel (Figure 6), we estimated the time taken for resistance prevalence of this antimicrobial to reach 50% in the future (Supplementary Figure 16). Concretely, we extracted the current resistance prevalence estimated at each pixel, and calculated the time difference from the current resistance prevalence (Supplementary Figure 16, time point a) until it reaches 50% (Supplementary Figure 16, time point b), using the corresponding regression models fitted in section “Trends of AMR for each antimicrobial class”.”

Supplementary Figure 16. Illustration of the estimation of the time it takes for resistance prevalence of an antimicrobial to reach 50%. Time point ‘a’ is associated with the estimated current resistance prevalence at a 10x10 km pixel; time point ‘b’ is associated with 50% resistance. The difference between ‘a’ and ‘b’ is the estimated time for resistance prevalence to exceed 50%.

I have some concerns about the dichotomization of prevalence of resistance to <50% and >50% (or 10%, 25%) as it would seem proximity to that threshold would be at least as important as patterns of resistance profiles in other locations in terms of informing the probability that the antibiotic in question would pass that threshold. These concerns are somewhat assuaged by the AUC cross validation but feel it at least deserves mention in limitations.

We agree with the reviewer that the dichotomization at 50% is in its nature subjective, and have added it as a limitation in lines 337 - 341:

“Fifthly, we dichotomized resistance prevalence using 50% threshold to define priority antimicrobials for AMR surveillance. We conducted sensitivity analysis by mapping priority antimicrobials using other thresholds (10% and 25%) as well. However, the choice of thresholds is dependent on multiple factors and in its nature subjective.”